# Transition between distinct hybrid skyrmion textures through their hexagonal-to-square crystal transformation in a polar magnet

Deepak Singh [1] ✉, Yukako Fujishiro [2], Satoru Hayami [3], Samuel H. Moody[1], Takuya Nomoto [4], Priya R. Baral[1], Victor Ukleev [5], Robert Cubitt [6], Nina-Juliane Steinke[6], Dariusz J. Gawryluk [7], Ekaterina Pomjakushina [7], Yoshichika Ōnuki[2], Ryotaro Arita [2,4], Yoshinori Tokura [2,8], Naoya Kanazawa [9] & Jonathan S. White [1] ✉

Magnetic skyrmions, topological vortex-like spin textures, garner significant interest due to their unique properties and potential applications in nanotechnology. While they typically form a hexagonal crystal with distinct internal magnetisation textures known as Bloch- or Néel-type, recent theories suggest the possibility for direct transitions between skyrmion crystals of different lattice structures and internal textures. To date however, experimental evidence for these potentially useful phenomena have remained scarce. Here, we discover the polar tetragonal magnet $EuNiGe_3$ to host two hybrid skyrmion phases, each with distinct internal textures characterised by anisotropic combinations of Bloch- and Néel-type windings. Variation of the magnetic field drives a direct transition between the two phases, with the modification of the hybrid texture concomitant with a hexagonal-to-square skyrmion crystal transformation. We explain these observations with a theory that includes the key ingredients of momentum-resolved Ruderman–Kittel–Kasuya–Yosida and Dzyaloshinskii-Moriya interactions that compete at the observed low symmetry magnetic skyrmion crystal wavevectors. Our findings underscore the potential of polar magnets with rich interaction schemes as promising for discovering new topological magnetic phases.

The study of topological phenomena arising from magnetically ordered spin systems that exhibit noncollinear and noncoplanar configurations has experienced rapid growth in recent years[1–4]. One notable example is the magnetic skyrmion[5–10], a nanoscale vortex-like structure that is characterised by swirling spins with an integer skyrmion winding number, $N_{Sk}$, defined as:

$$N_{Sk} = \frac{1}{4\pi} \int \mathbf{n} \cdot \left(\frac{\partial \mathbf{n}}{\partial x} \times \frac{\partial \mathbf{n}}{\partial y}\right) dx dy, \qquad (1)$$

[1]Laboratory for Neutron Scattering and Imaging (LNS), Paul Scherrer Institute (PSI), CH-5232 Villigen, Switzerland. [2]RIKEN Center for Emergent Matter Science (CEMS), Wako, Saitama 351-0198, Japan. [3]Graduate School of Science, Hokkaido University, Sapporo 060-0810, Japan. [4]Research Center for Advanced Science and Technology, University of Tokyo, Komaba, Meguro-ku, Tokyo 153-8904, Japan. [5]Helmholtz-Zentrum Berlin für Materialien und Energie, D-14109 Berlin, Germany. [6]Institut-Laue-Langevin, 6 rue Jules Horowitz, Grenoble 38000, France. [7]Laboratory for Multiscale Materials Experiments (LMX), Paul Scherrer Institut (PSI), CH-5232 Villigen PSI, Switzerland. [8]Department of Applied Physics, The University of Tokyo, Bunkyo, Tokyo 113-8656, Japan. [9]Institute of Industrial Science, The University of Tokyo, Meguro-ku, Tokyo 153-8505, Japan. ✉e-mail: deepak.singh@psi.ch; jonathan.white@psi.ch

which counts how many times the integrated solid angle between the neighbouring spin directions wrap a unit sphere. Here, $\mathbf{n}(\mathbf{r}) = \mathbf{m}(\mathbf{r})/|\mathbf{m}(\mathbf{r})|$ is the unit vector along the direction of the local magnetic moment $\mathbf{m}(\mathbf{r})$[4,5]. For an azimuthally symmetric skyrmion, one can write $\mathbf{n}(\mathbf{r}) = (\cos\Phi(\mathbf{r})\sin f(\mathbf{r}), \sin\Phi(\mathbf{r})\sin f(\mathbf{r}), \cos f(\mathbf{r}))$ in spherical coordinates, with position vectors $\mathbf{r} = r(\cos\phi, \sin\phi)$ in polar coordinates. Here $f(\mathbf{r})$ describes a polar angle, with $f(r=0) = \pi$ defining a skyrmion core moment alignment antiparallel with the magnetic field, and $f(r \to \infty) = 0$ defining moments parallel with the field at the skyrmion periphery. When this Ansatz is inserted into Eq. 1[4,5], one obtains a definition of the skyrmion vorticity $\omega$ from the solutions of $\Phi(\phi)$ alone, which for the considered boundary conditions $f(r=0) = \pi$ and $f(r \to \infty) = 0$ is an integer such that $N_{Sk} = -\omega$. Finally, if the moment azimuthal angle $\Phi$ varies smoothly with azimuthal direction $\phi$ of position $\mathbf{r}$, it follows that

$$\Phi(\phi) = \omega\phi + \chi \qquad (2)$$

where $\chi$ is defined as the skyrmion helicity. For a common vorticity $\omega = 1$, values of $\chi = \pm\frac{\pi}{2}$; 0 or $\pi$ respectively correspond to Bloch- and Néel-type skyrmions, these being the two skyrmion types observed in the majority of experimental research to date. Representations for these skyrmion archetypes are depicted in Fig. 1a, b; we illustrate the skyrmion winding by displaying its form on a unit sphere, as well as the

2D projections, which show how the moments rotate either in a plane perpendicular (Bloch-type) or parallel (Néel-type) to the radial direction of the skyrmion. The uniform colour of the spheres and projections indicates the typical scenario where spatial anisotropy in $\chi$ is absent around the skyrmion. However, compared with standard Bloch- and Néel skyrmions, it can be anticipated that helicity anisotropies may give rise to hybrid skyrmions, which exhibit an anisotropic combination of Bloch- and Néel textures.

Despite these expectations, the experimental realisation for hybrid skyrmions is challenging. A key requirement is a non-standard pattern of antisymmetric interactions in the system, e.g. Dzyaloshinskii-Moriya interactions (DMIs), which are usually critical for the stabilisation of magnetic skyrmion phases. Since skyrmions normally form a magnetic crystal that modulates along multiple directions simultaneously—a so-called multi-$Q$ structure[7,11–15]—the particular helicity of the magnetic texture imposed by the DMIs depends on the alignment of the magnetic propagation vectors $Q$ with respect to the crystal[6,16,17]. When $Q$-vectors align with high crystal symmetry directions, the general DMI textures expected in structurally chiral magnets with $T$ and $O$ point group symmetries favour Bloch-type skyrmions, while in magnets with $C_{nv}$ point groups Néel-type skyrmions are generally favoured. A departure from the Bloch and Néel archetypes is found in magnets of $D_{2d}$ or $S_4$ symmetry, wherein antiskyrmion spin textures with alternating Bloch and Néel-type helicities

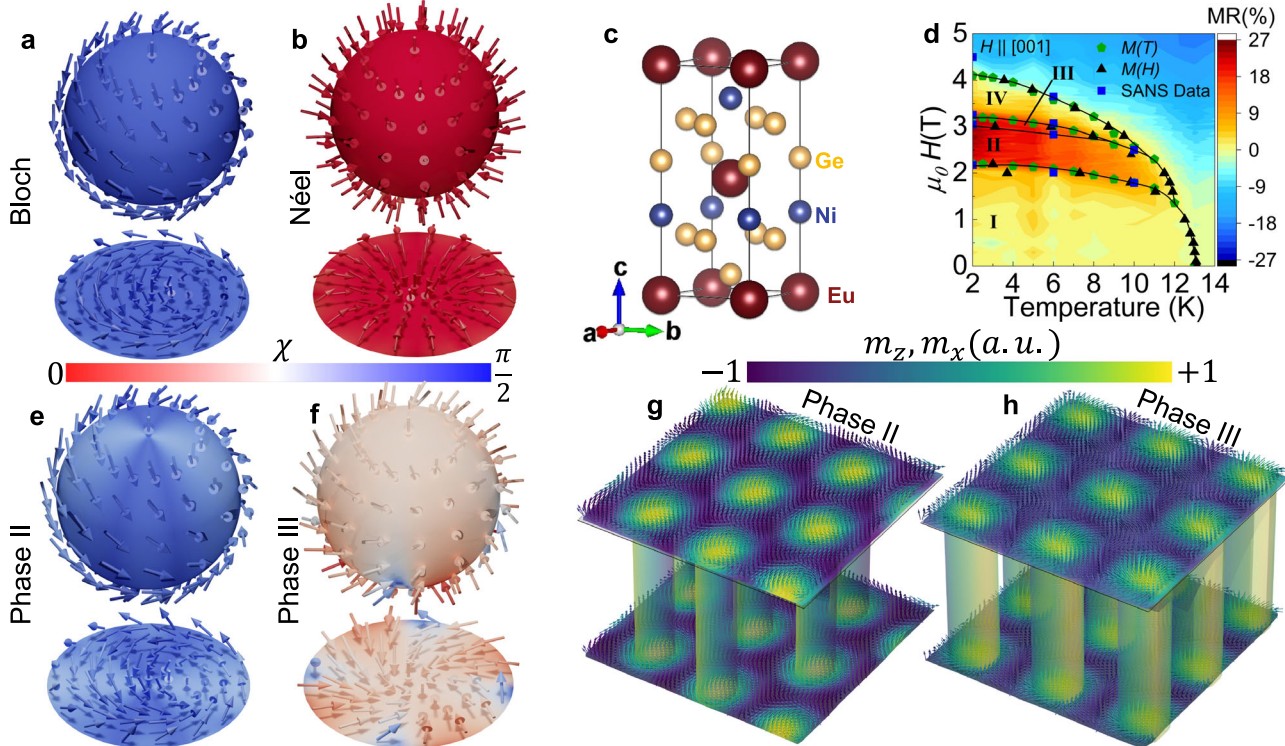

**Fig. 1 | Skyrmion helicities and skyrmion lattice transitions in EuNiGe₃.**
**a, b** Illustrations of archetype Bloch-type (**a**) and Néel-type (**b**) magnetic skyrmions with orthogonal helicity χ, depicted as both spherical wrappings and two-dimensional projections. The arrow glyphs indicate the local magnetic moment directions, $\mathbf{n}(\mathbf{r})$, whereas the colour map represents the angular helicity of the skyrmions, defined by $\chi = \arccos(\hat{\mathbf{n}}_\perp \cdot \hat{\mathbf{r}})$. Here, $\mathbf{n}_\perp$ is the normalised moment component perpendicular to the out-of-plane direction $\mathbf{c}$, given by $\mathbf{n}_\perp = \mathbf{n} - \hat{\mathbf{c}}(\hat{\mathbf{c}} \cdot \mathbf{n})$, and $\hat{\mathbf{r}}$ is the position relative to the skyrmion core. The north pole (south pole) of the sphere maps to the core (periphery) of the 2D skyrmion projection. **c** Polar tetragonal crystal structure of EuNiGe₃ which is of the BaNiSn₃-type (space group *I4mm*, No. 107, point symmetry $C_{4v}$) and exhibits broken inversion symmetry. **d** Magnetic phase diagram for *H* ∥ [001] derived from magnetisation, magnetoresistivity, and small-angle neutron scattering (SANS) measurements. As

seen in the phase diagram, there are four distinct phases, I, II, III, and IV. **e, f** The proposed skyrmion textures in phases II (**e**) and III (**f**) suggested from the results of our numerical simulations and polarised neutron scattering data. In phase II the anisotropy of the skyrmion texture is 'weakly' Bloch-type due to the existence of a minority Néel-type component. In phase III the skyrmion is weakly Néel-type close to the core, with a clearer in-plane anisotropy between Néel- and Bloch-type components at larger radial distances. **g, h** Three-dimensional schematic illustrations of the distorted hexagonal skyrmion crystal phase II (**g**) and square skyrmion crystal phase III (**h**) suggested by the simulation results and consistent with our experiments. The arrow glyphs represent the local moment direction and are colour-coded based on the component of magnetisation along the *c*-direction, $m_z$. The colouring of the skyrmion tubes displays the $m_x$-component of magnetisation, which is aligned with the tetragonal *a*-axis.

have been observed in thin-plate and micron-sized samples[18–21]. However, due to an apparent lack of observation of antiskyrmions in bulk samples, it remains an open question if antiskyrmions can be considered as an intrinsic bulk phenomenon; their generally large size (>100 nm) implicates their stability and internal texture as governed by an interplay between anisotropic DMIs and sample thickness-dependent dipolar interactions. In synthetic magnetic multilayer systems hallmarked by two-dimensionality and highly tunable magnetic interactions, various evidence[22–24] for hybrid spiral and skyrmion textures has been presented. Despite these examples, theoretical expectations[25,26], and further experimental studies in various reduced dimension settings[27–30], microscopic experimental evidence for a hybrid helicity of skyrmions as an intrinsic property of bulk magnets, and wherein sample thickness-dependent interactions play no role, has remained elusive.

One strategy to realise hybrid skyrmion formation in the bulk that has not been pursued to date, is to stabilise multi-$Q$ textures with magnetic $Q$-vectors aligned with reduced symmetry directions, so that interactions promoting both Bloch- and Néel-type helicities can each be simultaneously active and compete to define new spin textures. In the following, we exploit this strategy and show a polar tetragonal magnet EuNiGe$_3$ with $C_{4v}$ point symmetry (Fig. 1c) to be a remarkable host of two equilibrium hybrid skyrmion crystal phases. We construct the phase diagram (Fig. 1d) from magnetic, transport and neutron scattering measurements, and observe the material to display four distinct magnetic phases I to IV as a function of magnetic field below the Néel temperature. While the ground state phase I and highest-field phase IV display topologically trivial forms of incommensurate magnetic order, at intermediate fields we observe the direct magnetic field-induced transformation between a skyrmion crystal phase II (Fig. 1e and g), which displays a distorted hexagonal lattice structure, and phase III (Fig. 1f and h), which displays a square lattice structure. As seen in Fig. 1d, the adjacent hybrid skyrmion phases generate a strong enhancement of the magnetoresistivity. Microscopic evidence for an anisotropic combination of Bloch and Néel textures contributing to the hybrid skyrmion textures is provided by polarised neutron scattering, and backed up by numerical simulations. We explain both the stability of the hybrid skyrmion phases and the direct transition between them with an itinerant electron theory that invokes momentum-resolved exchange and antisymmetric Dzyaloshinskii-Moriya interactions that compete at the observed magnetic $Q$-vectors. The particular alignment of magnetic $Q$-vectors with the low symmetry directions of the polar lattice of EuNiGe$_3$ is thus found to be a critical ingredient for the realisation of hybrid skyrmions in the bulk.

## Results

To search for hybrid skyrmion textures, our target material is the $4f$-electron compound EuNiGe$_3$ with a polar tetragonal structure described by space group $I4mm$ (No. 107) (Fig. 1c). The magnetic properties are governed by Heisenberg-like Eu$^{2+}$ ($S = 7/2$, $L = 0$) ions, with both a spin density functional theory calculation and an earlier x-ray magnetic circular dichroism (XMCD) study[31] indicating negligible moments due to Ni and Ge (<1% of Eu). Previous work shows the system transitions to an incommensurately modulated helical order below the Néel temperature, $T_N = 13.2$ K[32]. The helical magnetic wavevector is confined within the tetragonal basal plane and found to be $\mathbf{Q} = (0.25, \delta, 0)$, $\delta \sim 0.05$ in reciprocal lattice units (r.l.u.)[33,34], and thus a low-symmetry direction of the polar lattice. The application of a magnetic field $H$ along the polar $c$-axis was found to induce several metamagnetic transitions[33–35]. However, details regarding the field-induced magnetic structures are either incomplete or missing[34]. Additionally, a non-monotonic $H$-dependence of the Hall resistivity was also found earlier and discussed in relation to topological Hall effects[35] but without connection to the proper picture of the underlying magnetic order. Finally, in a recent preprint, observation of phase II is reported[36];

however, phase III is overlooked, and the hybrid skyrmion helicity and crystal coordination transitions representing the main messages of our paper are neither observed experimentally nor described theoretically.

To determine the magnetic phase diagram of our single crystalline EuNiGe$_3$ samples, we performed bulk magnetic and transport measurements for magnetic fields applied along the polar $c$-axis (see Methods). Figure 2a, b respectively display typical $H$-dependent data at 2 K of the dc magnetisation ($M$) and the real part of the ac susceptibility ($\chi_{ac}'$) (see Supplementary Note I and Supplementary Fig. 1 for further details on the phase diagram determination by ac susceptibility). The $M$-$H$ and $\chi_{ac}'$-$H$ curves each show multiple step-like transitions prior to reaching the saturated state with $M = 7.85$ μ$_B$ / Eu$^{2+}$ at 6 T, and we identify four distinct phases I–IV. In both the $M$-$H$ and $\chi_{ac}'$-$H$ data, we observe hysteresis between $H$-increasing and $H$-decreasing sweeps, in addition to clear cusps in $\chi_{ac}'$, at the boundaries of phases II and III. These features infer both phases II and III to be equilibrium phases that each host a complex magnetic order.

From further $H$-dependent bulk measurements at higher temperatures, including neutron measurements discussed later, the $\mu_0 H$-$T$ phase diagram for $\mu_0 H \parallel c$ was constructed (Fig. 1d). The phase space occupied by both phases II and III coincides precisely with where we observe both a large enhancement of the magnetoresistance ratio (MR), defined as $[\rho_{xx}(\mu_0 H) - \rho_{xx}(0)]/\rho_{xx}(0)$ (see Methods). In addition, Fig. 2c shows anomalies in Hall resistivity $\rho_{yx}$ to accompany the $\mu_0 H$-dependent magnetic transitions. Although the non-linear, non-monotonic $\mu_0 H$-dependence of $\rho_{yx}$ evades a conventional interpretation (see Supplementary Notes II and III, and Supplementary Fig. 2 for the detailed discussion of the $\rho_{yx}$ data), the data nonetheless suggest an intimate relationship between magnetic order and electrical transport properties.

In this context, we also performed out-of-plane rotation measurements of the Hall resistivity as $H$ was rotated by an angle $\varphi$ with respect to the $c$-axis in the a–c plane. Here $I \parallel a$, and $\varphi = 0°$ corresponds to $\mu_0 H \parallel c$. Figure 2d shows the normalised Hall resistivity $\rho_{yx}(\varphi)/\rho_{yx}(\varphi = 0)$ obtained at 2 K, and magnetic fields appropriate for each phase. In phases I (1.5 T) and IV (3.5 T), $\rho_{yx}(\varphi)$ shows a near cosinusoidal variation with $\varphi$ and negligible hysteresis. This indicates $M$ to closely follow $H(\varphi)$, and that the main contributions to $\rho_{yx}$ are due to the normal and conventional anomalous Hall resistivities. In contrast, for phases II (2.6 T) and III (3.2 T), $\rho_{yx}(\varphi)$ displays abrupt anomalies as the field is tilted beyond $\varphi \sim \pm 10$-15° away from either $c$ or $-c$. These features denote the occurrence of first-order magnetic phase transitions, and are hallmarked by a hysteretic range of the response in $\varphi$ that spans up to ~30°. In analogy with the more pronounced hysteretic features in $\rho_{yx}(\varphi)$ observed in Gd$_2$PdSi$_3$[37], these data indicate each of the phases II and III in EuNiGe$_3$ to host spin textures with a distinct topology compared with phases I and IV.

To clarify the magnetic order in each phase we performed single crystal small-angle neutron scattering (SANS) measurements. SANS is generally sensitive to magnetic order if the Fourier transform of the local magnetisation density contains incommensurately modulating spin components $\hat{\mathbf{m}}(\mathbf{Q}) \exp[i\mathbf{Q} \cdot \mathbf{r}] + \text{c.c.}$ where components of $\hat{\mathbf{m}}(\mathbf{Q})$ are normal to $\mathbf{Q}$. Here $\hat{\mathbf{m}}(\mathbf{Q})$ is a complex vector and c.c. denotes complex conjugate. Figure 3a illustrates the transmission geometry of the experiment, where $H$ was applied parallel to both [001] and the neutron beam (see Methods for full details). Typical SANS data summarising the $H$-evolution of the magnetic order at 1.9 K during the $H$-increasing process are shown in Fig. 3b–h. Starting in the ground state phase I, the SANS data shown in Fig. 3e agree with previous works[33,34]. The pattern displays a four-fold symmetry due to the existence of four magnetic domains described by low symmetry $Q$-vectors equivalent to $\mathbf{Q}_1 = (0.25, \delta, 0)$, $\delta \sim 0.05$. Note that for each $Q$-vector, scattering appears at both $\pm Q$, leading to the eight reflections observed in total (Fig. 3i). From the observed magnitude of $|\mathbf{Q}_1| \sim 3.71$ nm$^{-1}$, we deduce the modulation period of ~1.69 nm. The earlier allocation of phase I as

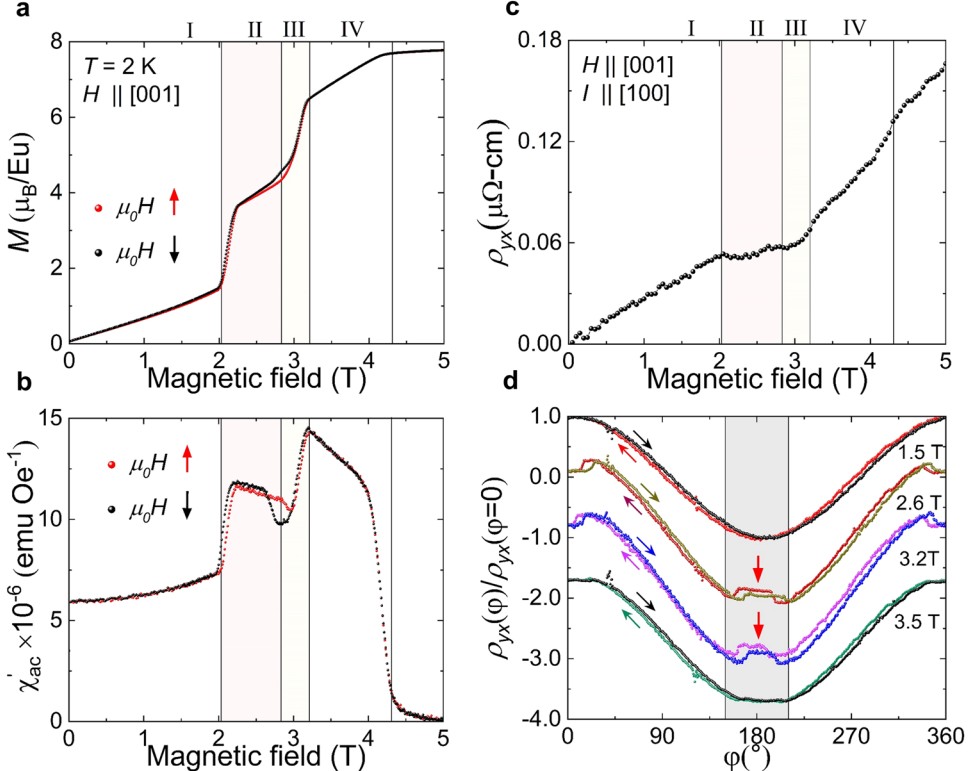

**Fig. 2 | Bulk magnetic measurements and transport data from EuNiGe₃.**
**a**, **b** respectively show the magnetic field-dependence of the bulk magnetisation $M$
(**a**) and ac susceptibility $\chi_{ac}'$ (**b**) at 2 K and $H \parallel [001]$. In both panels, red and black
symbols respectively correspond to the $H$-increasing and $H$-decreasing scans,
revealing hysteresis at the phase boundaries of phases II and III. **c** The magnetic
field-dependence of the transverse resistivity $\rho_{yx}$ at 2 K for $H \parallel [001]$ and $I \parallel [100]$. In
panels (**a**–**c**) shaded pink and yellow regions respectively denote the field stability

ranges of phases II and III. **d** Normalised transverse resistivity at 2 K with $H$ rotating
in the **a**–**c** plane. $H \parallel [001]$ corresponds to $\varphi = 0$. Data are obtained in phases I–IV, at
1.5 T, 2.6 T, 3.2 T and 3.5 T, respectively. The measurements at different fields are
offset along the vertical axis for clarity. The clockwise and counterclockwise $H$
rotation scans display hysteresis with a width of ~30° in phases II and III only. The
shaded grey region indicates the approximate range of hysteresis in $\varphi$ observed in
phases II and III.

being a multi-domain single-$Q$ helically-ordered phase[33,34] is broadly
consistent with the present SANS data, and our further SANS data
obtained in an in-plane magnetic field with $\mu_0 H \parallel [100]$ (see Supple-
mentary Note IV, and Supplementary Fig. 3).

From the ground state phase I, the $H$ evolution of the magnetic
order manifests in wholescale changes in the observed SANS patterns.
Figure 3f shows data obtained at 2.6 T that are typical for phase II.
Twelve diffraction spots are observed which may each be described by
one of three distinct types of $Q$-vector, $\mathbf{Q}_2 \sim (0.217, 0.065, 0)$,
$\mathbf{Q}_3 \sim (0.147, -0.147, 0)$, and $\mathbf{Q}_4 \sim (-0.065, -0.217, 0)$ (Fig. 3j). We inter-
pret the phase II SANS data to correspond to two equivalent domains
of a triple-$Q$ magnetic order that each break the four-fold symmetry,
and thus can be converted into one another by a 90° rotation around
the polar axis (Fig. 3j), with this allocation supported by further SANS
measurements done in magnetic fields tilted from the $c$-axis (see
Supplementary Note V, and Supplementary Fig. 4). Further increase of
$H$ leads directly to phase III which we find also hosts a multi-$Q$ struc-
ture. Figure 3g shows the typical SANS pattern for this phase at 3.2 T.
Sixteen Bragg spots are observed that can each be described by two
distinct types of short and long $Q$-vector. As shown in Fig. 3k, the
shorter $Q$-vectors include $\mathbf{Q}_5 \sim (0.15, -0.10, 0)$ and $\mathbf{Q}_6 \sim (0.10, 0.15, 0)$
and other equivalents. Mutually orthogonal wavevectors such as $\mathbf{Q}_5$
and $\mathbf{Q}_6$ describe a square-coordinated multi-$Q$ lattice, and two such
domains co-exist that are tilted from one another in the tetragonal
plane by ~25°. The longer $Q$-vectors, which include $\mathbf{Q}_7 \sim (0.25, 0.05, 0)$
and other equivalents, satisfy relations such as $\mathbf{Q}_7 = \mathbf{Q}_5 + \mathbf{Q}_6$. We thus
conclude these peaks amount to higher-order reflections, a telltale
signature of a multi-$Q$ structure[38–40]. Finally, in phase IV, a typical SANS
pattern obtained at 3.5 T is shown in Fig. 3h. These data can be

described similarly as phase I with $\mathbf{Q}_8 \sim (0.25, 0.05, 0)$ (Fig. 3l), and are
consistent with phase IV hosting multi-domain single-$Q$ magnetism.
While the data at hand do not allow us to rule out phase IV also being
multi-$Q$, e.g. double-$Q$, we note that features in our bulk data indicating
non-trivial spin textures are absent. In stark contrast, multi-$Q$ phases II
and III display clear features in both bulk and electrical transport
properties that are suggestive for topological spin textures, and
therefore that each hosts a form of magnetic skyrmion lattice.

A quantitative analysis of the SANS data summarised in Fig. 3b–d,
and Supplementary Fig. 5, demonstrate the first-order character of the
field-driven transitions between the different phases. Sharp $H$-depen-
dent changes are observed in SANS intensity when moving between
each phase (Fig. 3b), the magnitudes of the various $Q$-vectors (Fig. 3c),
and the skyrmion density $n_{Sk}$ relevant for phases II and III (Fig. 3d). In
phase II, since $|\mathbf{Q}_2| = |\mathbf{Q}_4| > |\mathbf{Q}_3|$, the triple-$Q$ skyrmion lattice is not
perfectly hexagonally coordinated; instead it is distorted in momen-
tum space along the direction orthogonal to $\mathbf{Q}_3$. The real-space mod-
ulation of this skyrmion lattice thus varies in the tetragonal plane from
1.92 nm due to $|\mathbf{Q}_2|$ and $|\mathbf{Q}_4|$, and 2.10 nm due to $|\mathbf{Q}_3|$. The reciprocal
lattice distortion can be parameterised by the axial ratio of an ellipse
that overlays the six SANS spots making up each single triple-$Q$ domain
(see Supplementary Note VI and Supplementary Fig. 6). We find the
ellipticity of the $Q$-vector distribution in phase II to vary from ~1.1 to
~1.15 as the field increases, leading to the direct implication that in real
space the skyrmions in phase II are elongated elliptically along the
direction of $\mathbf{Q}_3$. The regular square-lattice coordination in phase III
indicates the skyrmions in this phase to display a real-space modula-
tion of 2.30 nm. We note finally that phases II and III each display
skyrmion densities $n_{Sk}$ in the region of ~0.2-nm⁻² (Fig. 3d), which are

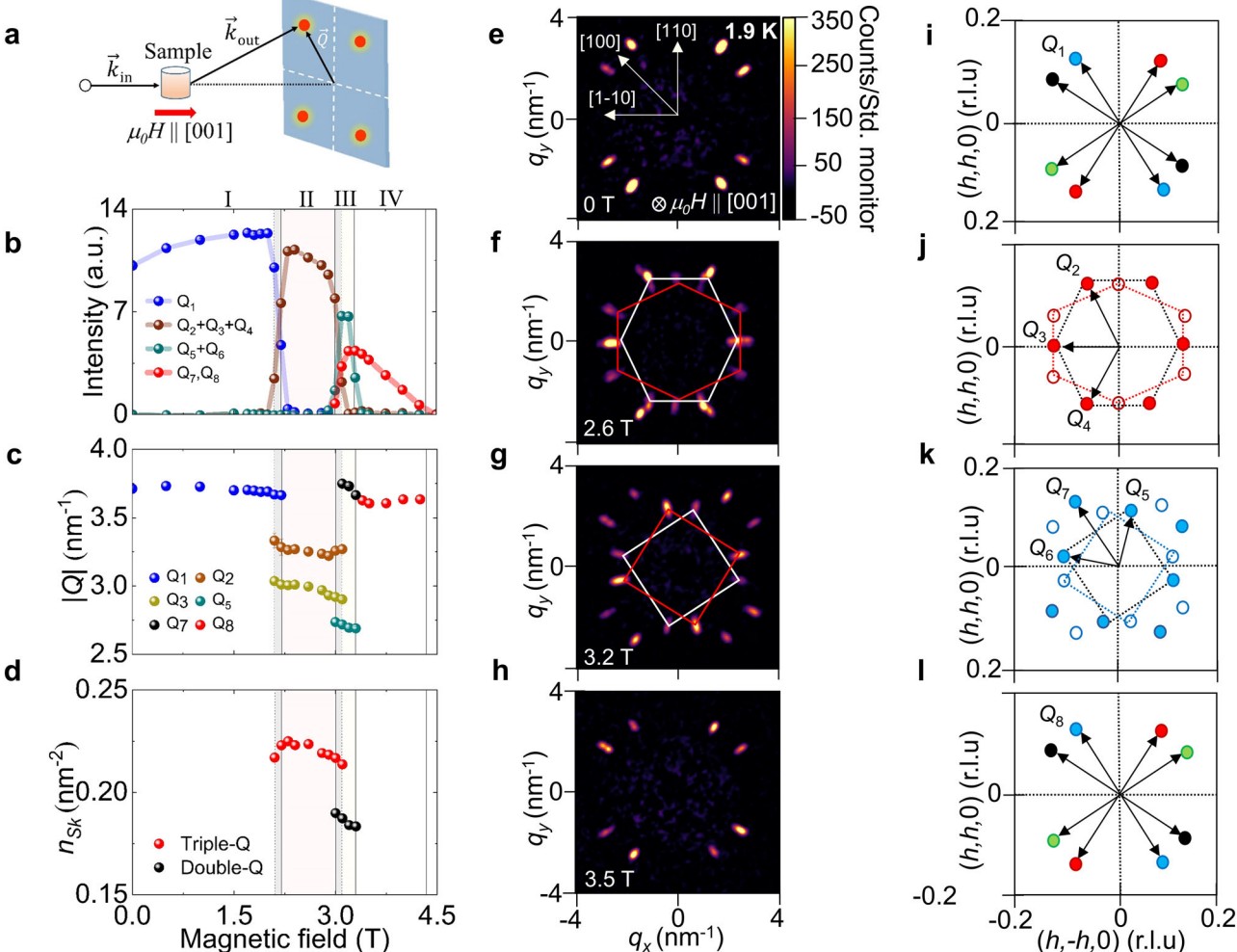

**Fig. 3 | Magnetic field dependent SANS data from EuNiGe₃. a** Schematic illustration of the experimental geometry for SANS measurement. $k_{in}$ and $k_{out}$ are the incident and scattered neutron wave vectors, respectively. **b**–**d** Magnetic field dependence for $\mu_0 H \parallel [001]$ at 1.9 K of the total diffracted SANS intensities from the magnetic structures in each phase (**b**), the lengths of the magnetic wavevectors $|Q_i|$ in each phase (**c**) where $i = 1$–7 as defined in panels (**i**–**l**) and the skyrmion density $n_{Sk}$ (**d**). Coloured lines are guides for the eye, while shaded pink and yellow regions respectively denote the field stability ranges of phases II and III. The shaded grey regions denote field ranges of observed hysteresis. **e**–**h** SANS data taken at 1.9 K in phases I to IV at various magnetic fields $\mu_0 H \parallel [001]$. The indicated intensity scale and crystal axes in panel (**e**) apply to all panels. The $Q$-vectors in all phases always lie

in the tetragonal plane, with no finite component measured along the z (i.e. polar axis). **i**–**l** Schematics showing the deduced reciprocal space distributions of the magnetic scattering observed in phases I to IV. The distributions for phases I (**i**) and IV (**l**) correspond to four domains of single-$Q$ order. Phase II is composed of two coexisting domains of a hexagonally coordinated multi-$Q$ order (**j**) while phase III hosts two coexisting domains of a square-coordinated multi-$Q$ structure, where higher-order modulation $Q_7 = Q_5 + Q_6$ (**k**). In combination with the bulk measurements, the SANS data reveal phases II and III each host multi-$Q$ magnetic skyrmion crystals. In panels **i** to **l**, different colour symbols distinguish the different single-$Q$ or multi-$Q$ domains present. In panels (**b**–**d**) error bars indicating the standard error are smaller than the symbol size.

amongst the highest for $4f$ electron magnets hosting skyrmions[15,37,38,41–43].

Finally, we employed a polarised SANS setup as illustrated in Fig. 4a to investigate the spin orientation of the magnetic order in phases I to III in more detail (see Methods for further experimental details). In the experiment the incident neutron spin polarisation was aligned either parallel ($P_i$) or antiparallel ($-P_i$) to the incoming beam, and also $H \parallel c$. The neutron spin orientation after the scattering process was analyzed, and the intensities of non-spin-flip (NSF) and spin-flip (SF) scattering processes measured separately at each Bragg peak. In accord with Fig. 4b, magnetic NSF scattering arises from components of $\hat{\mathbf{m}}(\mathbf{Q})$ parallel $P_i$ and normal to $Q$-vector, while SF scattering arises from components of $\hat{\mathbf{m}}(\mathbf{Q})$ simultaneously normal to both $P_i$ and $Q$[44]. These selection rules lead to straightforward expectations; for either a proper-screw helix (the components of $\hat{\mathbf{m}}(\mathbf{Q})$ all lie strictly perpendicular to $\mathbf{Q}$) or a pure Bloch skyrmion crystal phase, equal weights of NSF and SF scattering are expected at each $Q$-vector. For an easy-axis

cycloidal texture (the components of $\hat{\mathbf{m}}(\mathbf{Q})$ lie in the $Q$-$c$ plane) or a pure Néel-type skyrmion crystal, only NSF scattering is expected at each $Q$, with SF scattering entirely absent. As shown in Fig. 4c–g, and summarised in Table 1, each of the symmetry-distinct $Q$-vectors in phases I–III always show both NSF and SF scattering with varying weights. These data prove directly that the magnetic structures in each of the phases I to III include both in-plane and out-of-plane spin components normal to the $Q$-vectors, with the latter a notable requirement for any form of skyrmion phase.

The result that arises directly from the polarised SANS data is that the varying weights of NSF and SF scattering at the different $Q$-vectors rule out proper-screw helix and cycloidal spin textures for phase I, as well as conventional Bloch- and Néel-type multi-$Q$ skyrmion crystals for phases II and III. While unique determinations of the spin components $\hat{\mathbf{m}}(\mathbf{Q})$ cannot be achieved with the data at hand, the observed $Q$-dependent weights of NSF and SF scattering are a direct microscopic signature of anisotropy in the internal spin texture expected generally

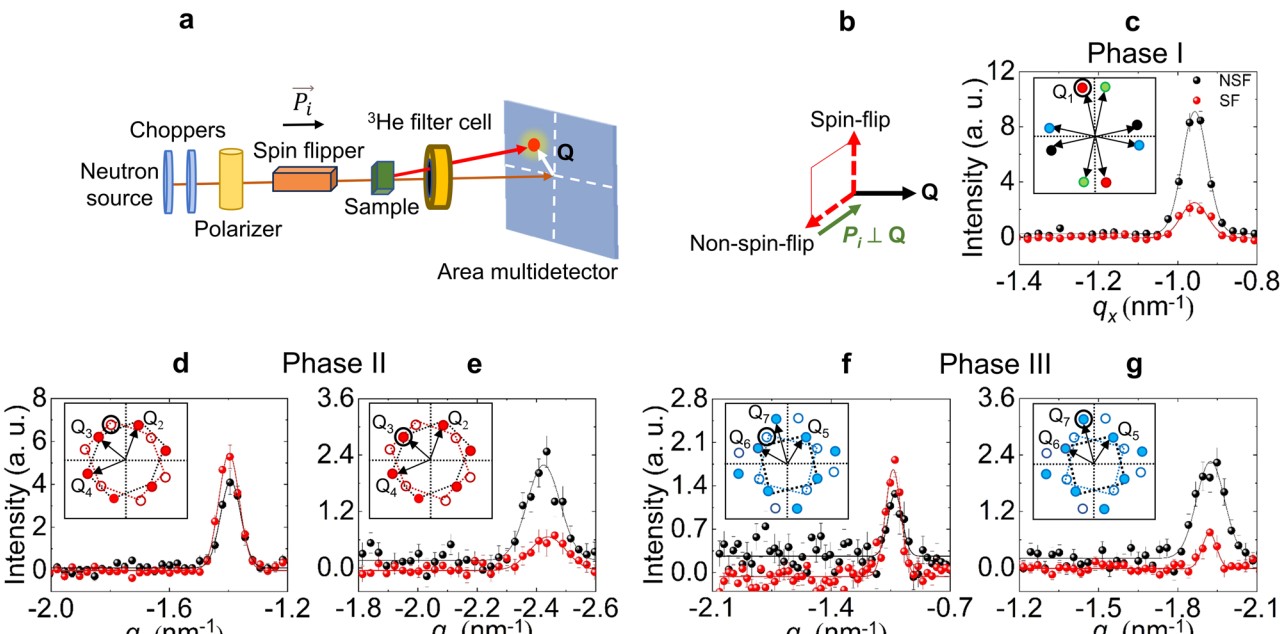

**Fig. 4 | Polarised SANS study for phases I, II, III. a** Schematic illustration of the polarised SANS experimental setup. **b** Schematic indicating non spin-flip (NSF) and spin-flip (SF) magnetic scattering cross-sections. $P_i$ represents the direction of the neutron polarisation, which can be aligned parallel or antiparallel with $k_i \parallel$ [001]. The red square denotes the plane containing $\hat{\mathbf{m}}(\mathbf{Q})$. **c–g** Line-scan profiles showing NSF (black symbols) and SF (red symbols) channels measured in phases I (**c**) II (**d, e**) and III (**f, g**), which are sensitive to the out-of-plane and in-plane components of $\hat{\mathbf{m}}(\mathbf{Q})$, respectively. The insets show schematic illustrations of the overall SANS patterns for phases I, II and III, with the diffraction peak measured in each sub-figure indicated by a circle. Data were obtained in phase I at 2 K and 0.5 T, in phase II at 2 K and 2.6 T, and in phase III at 6 K and 3.0 T. In panels (**c–g**) error bars indicate the statistical error.

for hybrid skyrmions in phases II and III. As also seen in Table 1, the data further provide a quantitative benchmark against which we compare the expectations of our theoretical analysis discussed later, and from which we suggest the reconstructed skyrmion textures for phases II and III shown in Fig. 1e, f.

Our main experimental observations that require theoretical explanation are firstly the origin of the magnetic $Q$-vectors observed in EuNiGe₃, secondly the direct transition between the skyrmion phases II and III, and finally the origin of the $Q$-dependent weights of NSF and SF polarised SANS scattering, since this provides the key insight for suggesting the anisotropy of the internal skyrmion textures.

First, we discuss the microscopic origin of the propagation vector $\mathbf{Q} = (1/4, \pm\delta, 0)$ in EuNiGe₃. By utilising spin density functional theory calculations, we derive a classical spin model composed of Eu spins through the local force method[45] (see Methods). Figure 5a presents a Fourier transform of the spin-spin interaction, $J(q)$, on the $q_z = 0$ plane. We observe two peaks at $\mathbf{Q} = (1, 0, 0)$ and $\mathbf{Q} \sim (1/4, 0, 0)$, both indicating the presence of an inherent spin modulation

### Table I | Experimental and Theoretical Ratios of NSF and SF polarised SANS intensities

| Phase | Q-vector type | Ratio NSF : SF Experiment | Ratio NSF : SF Simulations |
|---|---|---|---|
| I | $\mathbf{Q}_1$ | 1 : 0.3(1) | 1 : 0.8 |
| II | $\mathbf{Q}_2 (\equiv \mathbf{Q}_4)$ | 1 : 1.4(1) | 1 : 1.0 |
| II | $\mathbf{Q}_3$ | 1 : 0.4(1) | 1 : 0.7 |
| III | $\mathbf{Q}_5 (\equiv \mathbf{Q}_6)$ | 1 : 0.2(1) | 1 : 0.1 |
| III | $\mathbf{Q}_7$ | 1 : 1.9(3) | 1 : 1.9 |

Experimental data are determined from the integrated intensities of the peaks shown in Fig. 4c–g. These values are compared with those expected according to our numerical simulations. Results for phases I–III are given for the various symmetry-distinct Q-vector types defined in Fig. 3. The errors in the experimentally-determined NSF: SF ratios are the standard error.

instability in EuNiGe₃. Although the heights of these two peaks are comparable, we focus our discussion on the peak at $\mathbf{Q} \sim (1/4, 0, 0)$ as it appears to be a modulation vector close to that observed in the experiment. To investigate the origin of the $\mathbf{Q} \sim (1/4, 0, 0)$ peak, we decompose the total spin-spin interaction $J(q)$ into its orbital contributions due to Eu atoms[46]. From the orbital-decomposed $J(q)$, we find that the primary contribution arises from the Eu-5d orbitals, which is similar to Gd-based skyrmion compounds[46,47]. In particular, the Eu-$t_{2g}$ orbitals give rise to a peak at $\mathbf{Q} \sim (1/3, \pm\delta, 0)$ as depicted in Fig. 5b, which moves to $\mathbf{Q} \sim (1/4, 0, 0)$ by including the other sub-dominant contributions. Figure 5c illustrates the band structure of EuNiGe₃ calculated by assuming the paramagnetic state in the conventional unit cell. Although many bands intersect the Fermi level, we can see that band near the Γ-point in band 23 is predominantly formed by the $t_{2g}$ orbital. The corresponding Fermi surfaces are drawn in Fig. 5d. Interestingly, the aforementioned band dominated by the $t_{2g}$ orbital forms a square-box shape Fermi surface, which possesses a nesting vector $\mathbf{Q} \sim (1/3, 0, 0)$ as indicated by the black double-headed arrow. These results imply that this Fermi surface is responsible for the magnetic modulation observed in EuNiGe₃. It is noteworthy that the 5d orbital plays a crucial role in the magnetic modulation, like in the well-studied short-pitch skyrmion compound GdRu₂Si₂[38,46,48]. Therefore, we expect that the generalised Ruderman–Kittel–Kasuya–Yosida (RKKY) interaction, rather than the conventional RKKY interaction, acts as a driving force behind the formation of the observed skyrmion lattices.

Next, we turn towards obtaining an understanding for the $H$-driven transition between the skyrmion phases II and III, and the origin of their anisotropic internal magnetisation textures. Due to the non-centrosymmetric polar crystal structure and itinerant nature of the $4f$ magnetism in EuNiGe₃, RKKY interactions, DMIs, and higher-order exchange interactions can all in principle contribute to stabilising multi-$Q$ skyrmion phases. To understand our observations, we studied

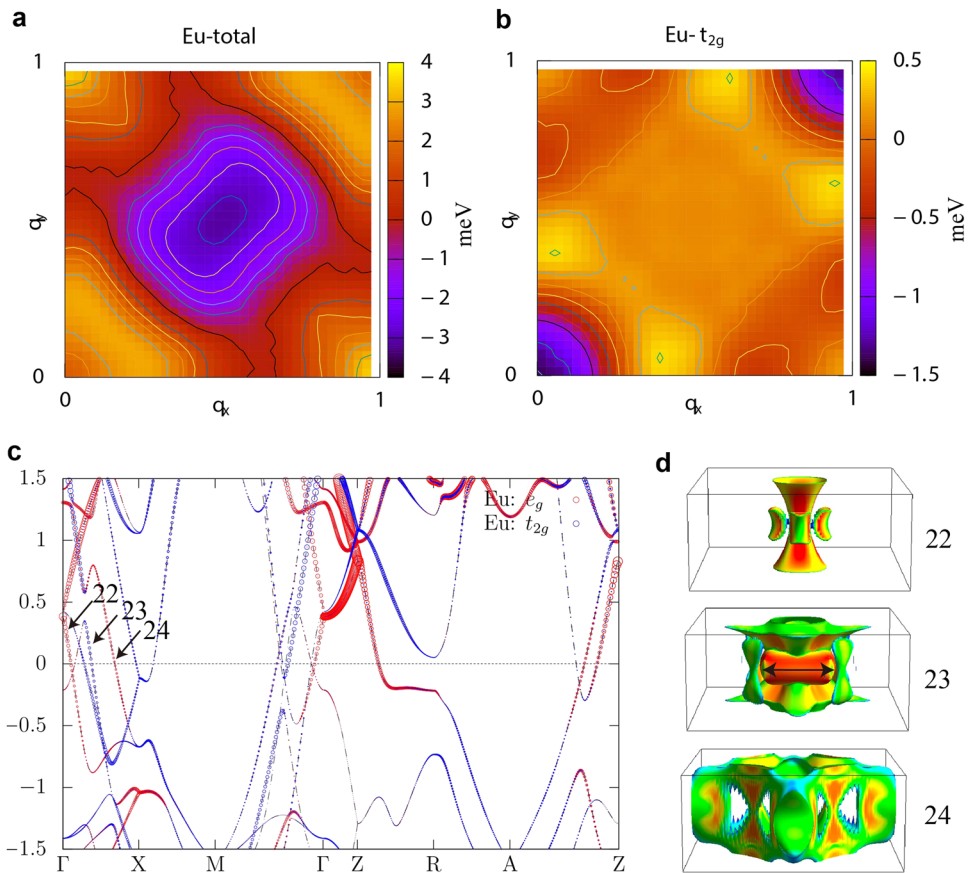

**Fig. 5 | Spin density functional theory (SDFT) calculations for EuNiGe₃. a** The spin-spin interaction $J(q_x,q_y,0)$ acting among Eu spins. **b** The Eu-$t_{2g}$ contribution to $J(q_x,q_y,0)$. **c** The band structure of the paramagnetic state in EuNiGe₃. **d** The Fermi surfaces of the paramagnetic state in EuNiGe₃. The colour represents the Fermi velocity.

the following classical Heisenberg model given by[49],

$$\mathcal{H} = \mathcal{H}_{ex} + \mathcal{H}_{DM} + \mathcal{H}_{Z} \tag{3}$$

where

$$\mathcal{H}_{ex} = -\sum_{\nu} J_{\mathbf{Q}_{\nu}} \left[ \left( S^x_{\mathbf{Q}_{\nu}} S^x_{-\mathbf{Q}_{\nu}} + S^y_{\mathbf{Q}_{\nu}} S^y_{-\mathbf{Q}_{\nu}} + \alpha_{\mathbf{Q}_{\nu}} S^z_{\mathbf{Q}_{\nu}} S^z_{-\mathbf{Q}_{\nu}} \right) \right.$$
$$+ \beta_{\mathbf{Q}_{\nu}} \left( S^x_{\mathbf{Q}_{\nu}} S^x_{-\mathbf{Q}_{\nu}} - S^y_{\mathbf{Q}_{\nu}} S^y_{-\mathbf{Q}_{\nu}} \right) + \gamma_{\mathbf{Q}_{\nu}} \left. \left( S^x_{\mathbf{Q}_{\nu}} S^y_{-\mathbf{Q}_{\nu}} + S^y_{\mathbf{Q}_{\nu}} S^x_{-\mathbf{Q}_{\nu}} \right) \right]$$

$$\mathcal{H}_{DM} = -\sum_{\nu} \mathbf{D}_{\mathbf{Q}_{\nu}} \cdot \left( \mathbf{S}_{\mathbf{Q}_{\nu}} \times \mathbf{S}_{-\mathbf{Q}_{\nu}} \right)$$

$$\mathcal{H}_{Z} = -H^z \sum_{i} S^z_i.$$

Here $\mathbf{S}_{\mathbf{Q}_{\nu}}$ is the Fourier transform of the classical localised spin $\mathbf{S}_i$ with wave vector $\mathbf{Q}_{\nu}$, where $\nu$ is the label of the dominant ordering vectors. Parameters $\alpha_{\mathbf{Q}_{\nu}}$, $\beta_{\mathbf{Q}_{\nu}}$ and $\gamma_{\mathbf{Q}_{\nu}}$ have their physical origin in the RKKY interaction and spin-orbit coupling as described in refs. 17,50, and thus in general depend on the electronic structure. The first term in Eq. 3 represents the symmetric exchange interaction and the second term corresponds to the antisymmetric DM-type exchange interaction. The third term is the Zeeman Hamiltonian, which introduces the effect of an external magnetic field. Contributions from higher-order exchange interactions are neglected.

To simplify the model, we extracted the dominant momentum-resolved interactions at the experimentally observed $Q$ points reported in Fig. 3 and Supplementary Fig. IV. We take into account

the interactions in momentum space by supposing global minima at, for example, $\mathbf{Q}_1$ and symmetry-equivalent wavevectors, for example, $\mathbf{Q}'_1 \perp \mathbf{Q}_1$. Phases I and IV are described by single-$Q$ spiral states, and their dominant interactions are determined by specific wave vectors along the directions $\pm \mathbf{Q}_1$. For the distorted hexagonal skyrmion lattice (SkL) in phase II, the relevant ordering vectors are $\mathbf{Q}_2$ to $\mathbf{Q}_4$. Accordingly, for the square SkL in phase III, the high symmetric wave vectors are $\mathbf{Q}_1$ ($\equiv \mathbf{Q}_7$) and $\mathbf{Q}'_1$. The shorter $Q$-vectors are defined as $\mathbf{Q}_5$ and $\mathbf{Q}_6$, so that $\mathbf{Q}_7 = \mathbf{Q}_5 + \mathbf{Q}_6$. We neglect the contributions from the other $Q$ components in the interactions for simplicity. We set the interactions to be $J_{\mathbf{Q}_1} = J_{\mathbf{Q}_2}/\kappa_1 = J_{\mathbf{Q}_3}/\kappa_1 = J_{\mathbf{Q}_4}/\kappa_1 = J_{\mathbf{Q}_5}/\kappa_2 = J_{\mathbf{Q}_6}/\kappa_2$, $D = |\mathbf{D}_{\mathbf{Q}_1}| = |\mathbf{D}_{\mathbf{Q}_5}|/\kappa_2 = |\mathbf{D}_{\mathbf{Q}_6}|/\kappa_2$, $D' = |\mathbf{D}_{\mathbf{Q}_3}|$ and $|\mathbf{D}_{\mathbf{Q}_2}| = |\mathbf{D}_{\mathbf{Q}_4}| = 0$; we also appropriately set the other parameters of ($\alpha_{\mathbf{Q}_{\nu}}, \beta_{\mathbf{Q}_{\nu}}, \gamma_{\mathbf{Q}_{\nu}}$) to reproduce the phase sequence observed in experiments. In particular, we introduce $\beta_{\mathbf{Q}_{\nu}}$ and $\gamma_{\mathbf{Q}_{\nu}}$ only for $\mathbf{Q}_2$ and $\mathbf{Q}_4$ as a key ingredient for deriving the hybrid hexagonal skyrmion crystal[50] (see Supplementary Note VII for full details of the parameters used in the simulation). Here, $\kappa_1$ and $\kappa_2$ describe the relative amplitudes of competing exchange and DMIs. Hereafter, we fix $J_{\mathbf{Q}} = 1$, $D = 0.2$, $D' = 0.03$ $\kappa_1$, $\kappa_1 = 0.935$, and $\kappa_2 = 0.85$. The model Hamiltonian described with Eq. 3 was minimized using the iterative simulated annealing procedure and a single-step Monte-Carlo dynamics with the Metropolis algorithm[49] (see Methods). The numerically obtained spin configurations are distinguished by spin- and chirality-related quantities described in Supplementary Note VIII.

Figure 6 shows the magnetic field-dependence of the stable spin textures resulting from the simulations for $H$ parallel to the polar axis. Consistent with the experimental results, we find four distinct modulated phases are stable before reaching saturation. Figure 6a–d show the simulated real-space spin textures deduced for each phase that are

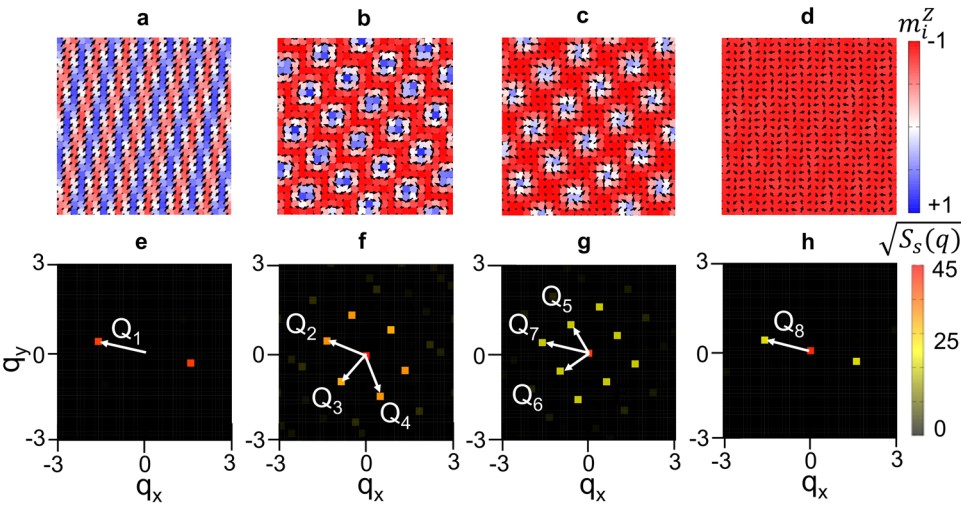

**Fig. 6 | Numerical simulation of magnetic structures for phase I, II, III, and IV.** **a**–**d** Real-space spin configurations of the single-$Q$ helical (**a**) triple-$Q$ hexagonal SkL (**b**) double-$Q$ square SkL (**c**) and single-$Q$ cone-like (**d**) spin textures. The contours show the $z$ component of the spin moment, while the arrows represent the $xy$ components of the spin moment. **e**, **h** Simulated SANS patterns for each phase, characterised by distinctive orientation of the fundamental magnetic modulation vectors. The stable spin textures are a single-$Q$ tilted-helical state (**e**) the distorted hexagonal SkL described by superposition of three magnetic modulations of $\mathbf{Q}_2$, $\mathbf{Q}_3$ and $\mathbf{Q}_4$ (**f**) the square double-$Q$ SkL state described by the superposition of two obliquely or orthogonally modulations of $\mathbf{Q}_5$, and $\mathbf{Q}_6$ (**g**) and a single-$Q$ cone-like (**h**) state.

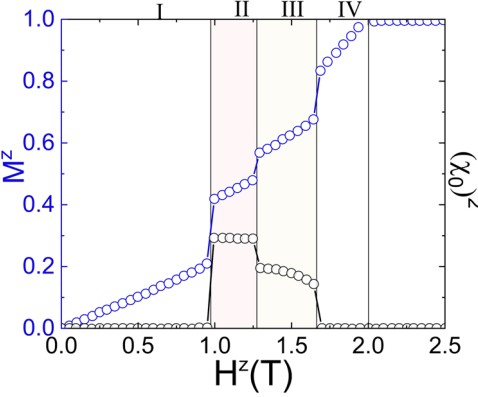

**Fig. 7 | Simulated magnetization and scalar spin chirality.** The magnetic field-dependence of the calculated uniform magnetisation, $M^z$, and squared spin scalar chirality, $(\chi_0)^2$ resulting from the simulations described in the main text. The shaded pink and yellow regions respectively denote the field stability ranges of phases consistent with those of phases II and III observed experimentally.

respectively consistent with phases I to IV, with the corresponding momentum space distribution of the spin structure factors shown in Fig. 6e to h. The stable spin textures for phases I and IV are found to correspond respectively to single-$Q$ tilted-helical and conical spiral phases, while those for phases II and III correspond to multi-$Q$ skyrmion lattices. The simulation further suggests the competing stabilisation mechanisms of skyrmion phases II and III. While the distorted-hexagonal phase II is stabilized by the interplay between the easy-axis interaction $\alpha_{\mathbf{Q}_v}$ and the symmetric anisotropic exchange interactions $\beta_{\mathbf{Q}_v}$ and $\gamma_{\mathbf{Q}_v}$, the square skyrmion phase III is stabilized by the DMI $\mathbf{D}_{\mathbf{Q}_v}$. Figure 7 shows the $H$-dependence of the magnetisation along the polar axis $M^z$ and the square of the scalar spin chirality $(\chi_0)^2$, with the latter becoming finite only in phases II and III, consistent with these phases hosting skyrmions.

In addition to reproducing the experimentally observed $H$-dependent sequence of phase transitions, the simulation results also provide insight for the origin of $Q$-dependent weights of NSF and SF polarised SANS scattering in skyrmion phases II and III, and ultimately

the hybrid internal texture of the skyrmions. The phenomenon for hybrid skyrmions has its roots in the assumption that the DMIs in the spin model can be described generally as a superposition of multiple types that favour different windings[17]. In the normal case, where spin textures modulate with high-symmetry $Q$-vectors, the DMIs are polar (namely perpendicular to both $Q$ and the polar axis) and result in a conventional Néel-type SkL. However, for spin textures with low-symmetry $Q$-vectors, such as in EuNiGe$_3$, the DMIs can be a superposition of polar and radial types, the latter favoring proper-screw spirals but different from pure chiral-type DMIs since its direction depends on the precise orientation of the wavevector.

According to our simulations, in phase II, the presence of superposed polar and radial DMIs at $\mathbf{Q}_2$ and $\mathbf{Q}_4$ hinders the formation of a conventional hexagonal Néel-type SkL. Simultaneously, symmetric anisotropic exchange interactions arising from $\beta_{\mathbf{Q}_v}$ and $\gamma_{\mathbf{Q}_v}$ favour the Bloch-type SkL[50]. This leads to competition between the Néel-type and Bloch-type SkLs, resulting in an anisotropic texture combining the two types. The square SkL phase III is also a hybrid of Néel and Bloch types due to a helicity competition caused by the superposition of different DMI types, without contributions from symmetric anisotropic exchange interaction. By analysing the stable spin textures expected from the simulations, we obtained the ratio of NSF to SF scattering expected at each $Q$-vector which can be compared directly with the polarised SANS data. As shown in Table 1, the model expectations capture the experimental result generally well, particularly in phases II and III. While fine-tuning of the model parameters and/or consideration of further interactions may further improve the agreement between the theory and the experimental data, our main insight remains valid in general; hybrid skyrmion crystals can arise in the presence of a $Q$-dependent competition between interactions favouring different helicity spin textures.

## Discussion

In Fig. 1e, f, we visualise the skyrmion textures in phases II and III resulting from the simulations. The hybrid aspect of the internal textures is visible in the contrast between the spatial distributions of the calculated helicity parameter χ of phases II and III, and those expected for pure Bloch (Fig. 1a) and Néel (Fig. 1b) skyrmions. From a quantitative analysis of the calculated χ values for each skyrmion type (see

Supplementary Note IX for the full details), we describe the skyrmions in phase II to show a 'weak' Bloch-type winding, with certain directions related to $Q_2$ and $Q_4$ showing a stronger Bloch character than the direction related to $Q_3$. Phase III is hallmarked by a generally weak Néel character near the core, with a noticeable in-plane azimuthal variation between Bloch and Néel character at larger radial distances. Since conduction electrons can couple to noncoplanar spin structures through the Berry curvature, the anisotropy of spin noncoplanarity displayed by hybrid skyrmion structures should be detectable in various emergent electrodynamic responses[51], particularly when the direction of current flow is varied in the tetragonal plane. Although the coupling between the conduction electrons and the magnetic structure appears to be weak in EuNiGe$_3$ (see Supplementary Note III), our results suggest a possible pathway to finding unique transport responses in other hybrid skyrmion host systems with a stronger coupling.

A second unique aspect of our observations is the direct transition between hexagonal and square equilibrium skyrmion crystal phases, which previously in bulk materials was observed only under far-from-equilibrium conditions[52,53]. Visualisations of the skyrmion crystal phases we observe in EuNiGe$_3$ are shown in Fig. 1g, h. Apart from EuAl$_4$, which displays a less drastic transition between double-$Q$ square and rhombic skyrmion phases with slightly different mutual orientations of the fundamental $Q$-vectors[42], the majority of $4f$-electron skyrmion systems host a single skyrmion phase with a lattice structure that reflects the underlying crystal symmetry[37,38,41,43]. In EuNiGe$_3$, our theoretical analysis indicates the stability of the ground state magnetic order with low symmetry $Q$-vectors as most likely determined by the Fermi surface structure, while the $H$-tuned competition between various competing interactions at the skyrmion crystal $Q$-vectors is responsible for the transition between skyrmion crystals that each spontaneously and differently break the crystal symmetries. The present study thus provides not only a demonstration of an externally driven and drastic reconfiguration of equilibrium skyrmion crystals hitherto unique to bulk magnets, it also hints at the associated materials principles and magnetic interaction landscapes that can both support and provide control over such phenomena. Possible ways forward for the latter include precise tuning of the chemical composition, or uniaxial strain engineering. Finally, the effects we have uncovered experimentally may be exploited in principle for various applications, for example, those involving a dynamic reconfiguration of skyrmion-based magnonic crystals.

## Methods
### Crystal growth and characterisation
Single crystals of EuNiGe$_3$ were grown by the In-flux method. The phase purity and crystallographic parameters of single crystal samples were confirmed by single crystal X-ray diffraction at room temperature using a STOE STADIVARI four-circle diffractometer equipped with a Dectris EIGER 1 M 2 R CdTe detector and incoming micro-focused Ag Kα radiation of wavelength 0.56083 Å. Data reduction and correction were performed using the X-Area package[54]. Solution, refinement, and absorption correction were done utilizing the JANA 2020 software (version 1.3.51)[55]. Good agreement between measured and refined intensities of over 331 reflections (184 unique) confirmed our crystals to display the nominal occupancy of the different atoms within uncertainty, and tetragonal lattice parameters of $a = 4.341(1)$ Å and $c = 9.896(4)$ Å, consistent with those reported previously[32–34]. Since our bulk magnetic measurements on the crystals reveal characteristic transitions to take place at temperatures and magnetic fields consistent with those reported previously[32–35], we conclude our crystals to be comparable in terms of quality to those studied in earlier work. For neutron scattering and transport measurements, crystal orientations were determined by Laue X-ray diffraction, and then the samples were cut out from the crystal with a wire saw and mechanically polished.

### Magnetisation, ac susceptibility and transport measurements
Magnetisation was measured using a SQUID magnetometer (Magnetic Property Measurement System, Quantum Design). Ac susceptibility measurements were performed using commercial magnetometers from Quantum Design (QD). The excitation field was kept constant at 1 Oe, while the two excitation frequencies were used for all measurements, 911 Hz and 917 Hz. The longitudinal ($\rho_{xx}$) and Hall ($\rho_{yx}$) resistivities were measured using the resistivity option or ac-transport option in a physical property measurement system (PPMS, Quantum Design). Measurements were performed on crystls with dimensions of 1 mm × 0.9 mm × 0.5 mm (for magnetisation and longitudinal resistivity) and 1 mm × 0.5 mm × 0.1 mm (for Hall resistivity), with the widest surface parallel to the (001) plane. For the transport measurements with out-of-plane magnetic field rotation, the single crystal was cut into a lamella with a thickness of about 500 nm to ensure sufficient current density and data quality by using focused-ion-beam (FIB) machining. The largest surface was parallel to the (001) plane while the electric current was applied along the [100] axis.

### Small-angle neutron scattering (SANS) measurements
Unpolarised SANS measurements were carried out using the SANS-I instrument at the Swiss Spallation Neutron Source (SINQ), Paul Scherrer Institute, Switzerland. The SANS data were obtained from ~5 mg single crystals of typical volume 2 × 2 × 0.15 mm$^3$. The widest (001) plane was aligned initially along the neutron beam. The crystal was mounted inside a superconducting cryomagnet providing temperatures down to 2 K, and magnetic fields up to 6.8 T along the $c$-axis. The neutron wavelength was selected to be 3.1 Å (Δλ/λ = 17 %). The incoming beam was collimated over 3 m before the sample, and scattered neutrons collected by a multi-detector placed 1.85 m behind the sample. The momentum-space resolution in the region of the observed $Q$-vectors was ~ 0.06 Å$^{-1}$. The SANS diffraction patterns like those in Fig. 3 were obtained by summing together two-dimensional multi-detector measurements taken over a focused range of sample + cryomagnet rotation (rocking) angles close the Bragg conditions for the various diffraction peaks. Background data acquired above $T_N$ was subtracted from the data at low temperature to leave only the magnetic signal. All magnetic field scans were conducted after zero-field cooling (ZFC) and during an $H$-increasing process.

Polarised SANS measurements on the same samples were carried out using the D33 instrument at the Insitut Laue-Langevin (ILL), France[56]. The instrument was operated in a time-of-flight mode that passed a wavelength band of 2–14 Å neutrons. A supermirror polariser was use to produce a polarised neutron beam with neutron spins aligned transverse to $k_i$. A longitudinally polarised beam was obtained by using guide fields, with the neutron spin polarisation at the sample position directed to be either parallel or antiparallel to the $c$-axis by using a spin flipper. The sample was positioned inside a horizontal-field magnet that could apply a maximum magnetic field up to 3.0 T (thus precluding the study of phase IV). A cryostat insert provided sample temperatures down to 2 K. A nuclear spin-polarised $^3$He analyzer cell was placed between the sample and the detector. Its polarisation direction could be also flipped to be either parallel or antiparallel to the direction of the incoming beam polarisation, and thus allow distinction between SF and NSF scattering signals. The magnetic scattering at specific $Q$-vectors in phases I to III was observable at specific scattering angles and selected neutron wavelengths within the 3-4 Å portion of the wavelength spectrum with Δλ/λ ~ 10 %. The incoming beam was collimated over a distance of 7.8 m before the sample, and scattered neutrons collected by a multi-detector placed 1.9 m behind the sample. In the defined wavelength band, the momentum-space resolution in the region of the observed $Q$-vectors was ~0.03 Å$^{-1}$. The polarising efficiency of the setup, characterised by the flipping ratio, was measured and found to vary from 6-8 depending on the wavelength and $H$. By measuring all possible neutron-spin-state

combinations of the unscattered beam, fully quantitative corrections for imperfections of the overall polarising efficiency are applied in the data analysis. All unpolarised and polarised SANS data analysis was carried out using GRASP software[57].

## Theoretical calculations

Spin density functional theory (SDFT) calculations for $EuNiGe_3$ were performed based on the projector augmented wave scheme[58] implemented in the Vienna ab initio simulation package[59,60]. The lattice constants and the atomic positions were taken from the experiment[61]. The exchange correlation functionals proposed by Perdew-Burke-Ernzerhof [62], $E_{cut} = 450$ eV as the cutoff energy for the plain wave basis set, and $N_k = 10 \times 10 \times 10$ as the number of $k$-points were used in the self-consistent field calculations. For the calculations of the paramagnetic state, we used the open core potential with $4f^7$ configuration for Eu element. To illustrate the Fermi surfaces with dense $k$-grid, we used the Wannier interpolation technique by using the WANNIER90 package[63,64]. Here, 182 Bloch bands on the $8 \times 8 \times 8$ $k$-grid were taken into account to obtain a 52-orbitals tight-binding model, where the Eu-$d$, Ni-$s$, $p$, $d$ and Ge-$s$, $p$ orbitals were used as the trial localised orbitals. Based on the obtained tight-binding model, the Fermi surfaces are drawn by FermiSurfer package[65] with the eigenvalues evaluated on $96 \times 96 \times 96$ $k$-grid. For the calculations of spin-spin interaction $J(q)$, we used the Wannier interpolation for the ferromagnetic solution of the SDFT + U calculations, where $U = 6.7$ eV and $J = 0.7$ eV were added into the Eu-$4f$ orbital. Here, 184 Bloch bands on the $8 \times 8 \times 8$ $k$-grid were used to obtain a 66-orbitals tight-binding model, where the trial orbitals of Eu-$d$, $f$, Ni-$s$, $p$, $d$ and Ge-$s$, $p$ were used. The total and orbital resolved spin-spin interactions $J(q)$ were then calculated by the local force method as implemented in refs. 45–47. Here, $N_k = 36 \times 36 \times 36$ as the number of $k$-points and $T = 20$ K were used in the calculations.

The numerical simulations of the magnetic phase stabilities were done on a classical spin model with contributions from symmetric exchange, DMI, and Zeeman energies. The magnetic energy density has been minimised using the iterative simulated annealing procedure and single-step Monte-Carlo dynamics with the Metropolis algorithm. We imposed periodic boundary conditions and performed simulations for lattices of different sizes to check the stability of the numerical routine.

## Data availability

The data presented in the figures that support the findings of this study are available either at the Zenodo online repository with identifier https://doi.org/10.5281/zenodo.10222231, or from the corresponding authors upon reasonable request. The raw neutron scattering data obtained using the D33 instrument at the ILL can be obtained at https://doi.org/10.5291/ILL-DATA.5-41-1202.

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

## Acknowledgements

We thank I. Plokhikh for support with single crystal X-ray diffraction characterisation of the samples studied in this work. This work is based partly on experiments performed at the Swiss Spallation Neutron source SINQ, Paul Scherrer Institute, Villigen, Switzerland and the Institute Laue-Langevin, Grenoble, France. We thank the Crystal Growth Facility (EPFL), the Laboratory for Mesoscopic Systems (ETH Zürich) and the Laboratory for Multiscale Materials Experiments (PSI) for providing access to the magnetometers. We acknowledge funding from the Swiss National Science Foundation (SNSF) through project grant 200021_188707 (D.S., V.U., S.H.M., P.R.B., J.S.W.), 200020_182536 (P.R.B.) and the Sinergia Network "NanoSkyrmionics" grant no. CRSII5_171003 (V.U., J.S.W.). Y.F. acknowledges funding from a JSPS Grants-in-Aid for Scientific Research Grant No. 22K14011. S.H. acknowledges funding from JSPS KAKENHI Grants Numbers JP21H01037, JP22H04468, JP22H00101, JP23H04869 and JST PRESTO Grant Number JPMJPR20L8. T.N. acknowledges support from JST PRESTO Grant Number JPMJPR20L7. R.A. acknowledges funding from JSPS KAKENHI Grants Numbers 19H05825 and 21H04437. N.K. and J.S.W. acknowledge financial support from the ETH Zürich Research Partnership Grant RPG 072021_07.

## Author contributions

The project was conceived by D.S., Y.F., V.U., Y.T., N.K. and J.S.W. Single crystals of EuNiGe$_3$ were prepared by D.S. and Y.Ō. D.S., Y.F. and P.R.B. performed magnetisation, ac susceptibility and magnetoresistance measurements. Sample preparation and characterisation work at PSI was supported by D.J.G. and E.P. Y.F. prepared the thin-plate lamella samples and performed the out-of-plane rotation resistivity measurements with support from Y.T. and N.K. Unpolarised and polarised SANS measurements were performed and analyzed by D.S., S.H.M., R.C., N.-J.S. and J.S.W. T.N. and R.A. performed the first-principles theoretical calculations. S.H. developed the effective spin Hamiltonian and performed numerical simulations. D.S., S.H. and J.S.W. prepared the manuscript, with further contributions from Y.F., S.H.M., T.N., and P.R.B. The results were discussed and interpreted by all the authors. The project was supervised by N.K. and J.S.W.

## Competing interests

The authors declare no competing interests.
