## [Peer Review File · Nature Communications]

REVIEWER COMMENTS

Reviewer #1 (Remarks to the Author):

D. Singh et al. investigated the magnetic field dependent magnetic properties of an itinerant polar magnet EuNiGe_3 , via a combination of experimental techniques, including magnetization, magneto-transport and neutron scattering. A magnetic field induced transition between hexagonal and square crystals of hybrid skyrmions is observed. A theory that invokes Heisenberg exchange and Dzyaloshinskii-Moriya interactions (DMI) is proposed to explain the experimental findings.

Although some experimental works had been done before in EuNiGe_3 using similar techniques, this study presents a more complete investigation of its magnetic field dependent properties. Particularly, the magnetic field induced transition between hexagonal and squares crystals of magnetic skyrmions is interesting. Also, the theoretical reasoning for this observation can be applicable to a broad class of magnetic materials. Considering the significance, this work has the potential of being accepted for publication at Nature Communications. That being said, some concerns need to be addressed before the publication of this manuscript.

1) What are the valence states of Ni and Ge atoms in EuNiGe_3 ? Do they also have magnetic moments that may alter the picture presented in this work?

2) Magnetic field dependent evolution of the propagation vectors is measured via the small angle neutron scattering (SANS) technique. However, it is not clear whether there is any non-zero contribution in the z-component of these vectors, especially the case when magnetic field is turned on in phase regions II-III. This concern need to be discussed.

3) Lines 32-34: the authors present a theory that invokes multiple exchange and DMIs to explain the field induced transition between hexagonal and square skyrmion lattices. But the key information in the manuscript is not reflected from this sentence. Heisenberg exchange couplings and DMIs are commonly present in lots of magnetic materials, either insulating or metallic.

Firstly, “multiple exchange” is a very vague term to describe Heisenberg exchange couplings, either isotropic or anisotropic, from different origins.

Secondly, the authors argue that the generalized RKKY interaction is important for the stabilization of the observed skyrmion lattices. In addition, the authors argue that the low symmetry Q vectors are

important for the observation of magnetic field induced transition between hexagonal and square skyrmion lattices. It seems that these key ingredients are not specified in the abstract.

Lines 272-273: On a related note, the authors need to be specific about the different parameters (α_Q , β_Q , γ_Q) presented in the Heisenberg exchange model and discuss the origins for these terms. For instance, does spin-orbit coupling play a role here?

4) The experimental resolution of SANS and polarized SANS need to be mentioned. The corresponding errors of the experimental data need to be included as well.

Some additional comments for this manuscript:

1) Some figures presented in the current version of this manuscript are of very low quality. Also, the font size in some figures should be larger and consistent with the other figures (such as Fig 1d and fig 6). Particularly, figs. 6a-h are too small compared to other figures, such as fig. 5.

2) Lines 46-52: some symbols introduced here are not well defined. Some general readers may find them confusing.

3) Line 171: how is that a 90 deg rotation?

4) Line 207-215: for the polarized SANS, how will the non-perfect spin-flip and non-spin-flip affect its ratio?

5) Line 325-338: it appears to me that there exists some competition rather than "frustration".

6) Line 379, powder x-ray diffraction is not a direct method to confirm the sample purity. Citing an early work or presenting with more data will help.

7) Line 670, Fig 2. χ_{ac} was used to describe the ac susceptibility. This should be applied to the main text as well.

8) Fig 3 and table I: multiple magnetic Q vectors are defined when describing its magnetic field dependence. Some readers may get confused. The authors may consider updating the caption and the table with more details, such as $Q_1 \equiv Q_7$ and $Q_7 = Q_5 + Q_6$.

9) Fig 6 and Supplementary Note VII: the authors may consider adding a table including all the parameters used for the simulations in Fig 6 under different magnetic fields.

To summarize, the experimental observations and theoretical arguments in this work are interesting and can have a broad impact to the community focusing on magnetic skyrmions. But some concerns need to be addressed and some key information need to be highlighted before the acceptance for publication.

Reviewer #2 (Remarks to the Author):

In the present manuscript, the authors investigate a direct transition between hexagonal and square crystals of hybrid skyrmions in a polar magnet EuNiGe_3 . To prove the concept, the authors utilize versatile experimental techniques such as bulk magnetic and transport measurements, measurements of the Hall resistivity, SANS, susceptibility measurements etc. The authors convincingly prove that the ultra-small skyrmions change their helicity from Bloch to Neel fashion by an applied magnetic field.

The paper is certainly of great use for skyrmionics and can revolutionize various applications of skyrmions in particular involving a dynamic reconfiguration of skyrmion-based magnonic crystals. Moreover, the paper is written by a pioneer of skyrmionics Prof. Y. Tokura who discovered skyrmions and introduced the modern language of skyrmionics. In particular, in Ref. [5], Prof. Tokura introduced the notion of the topological charge, which allowed to classify skyrmions and anti-skyrmions. The manuscript is well-written and clear and I recommend it to publication.

In the following, I just would like to raise some points, which could be amended.

1. The motivation of the manuscript is not quite clear to me. Is it a phase transition between hexagonal and square skyrmion lattices or the change of skyrmion helicity during this phase transition? Unfortunately, after reading the manuscript, I cannot clearly express what is the main idea of the paper?

A. If the motivation of the paper is the square or the hexagonal alignment of skyrmions, then, one can recall some papers, in particular [T. Nakajima, H. Oike, A. Kikkawa, E. P. Gilbert, N. Booth, K. Kakurai, Y. Taguchi, Y. Tokura, F. Kagawa, and T. Arima, Skyrmion lattice structural transition in MnSi , *Sci. Adv.* 3, e1602562 (2017).]. The authors observed the transition from the hexagonal to the square skyrmion order. In this sense, there are many papers with the same motivation.

B. If the motivation of the paper is the value or change of the skyrmion helicity between 0 and $\pi/2$, which differs from the Bloch or Neel fashion of rotation, then, one would first recall the chiral magnets

with C_n symmetry. In particular, in Ref. [Chem. Rev. 2021, 121, 5, 2857–2897], Prof. Tokura explicitly introduced the Lifshitz invariants for this point-group symmetry in Eq. (9). So, I wonder why this symmetry class is not even mentioned in the present manuscript?

C. Then, one could recall frustrated skyrmions, which change their helicity or ultra-small skyrmions in monolayers with induced DMI etc.

D. One can also recall the change of skyrmion helicity near the surfaces of thin-film helimagnets, the so-called surface twists.

2. I really apologize for my complaint but the resolution of figures 1 and 6 is so low that it is impossible to discern the structure of different spin distributions. Indeed, the authors speculate that the helicity of skyrmions changes, but eventually it is impossible to see any spin structure, the information is hidden. I wonder why?

A. I would recommend to show the structure of all phases. I believe that one image would immediately clarify the point raised. In particular, is it possible to see the structure of a spiral state with the wave vector rotating in-plane perpendicular to the field?

B. Then, I would recommend to describe in more detail the method how the helicity was defined. If I understand correctly, for very small skyrmions when the angle between adjacent spins is large (BTW, what is the angle?), one should first determine the skyrmion center. After that, one should measure the helicity along the radial direction. Does the helicity changes along the radial coordinate or it remains constant?

C. It can be the case that for such small skyrmions and periodic boundary conditions applied, one may numerically obtain different solutions. For example, in one solution, a skyrmion center may be located at the grid knot, in another – between the points of the numerical grid. I wonder, how the authors control this process? Could the authors show their solutions or this information is also hidden?

3. A related question: the authors introduce the notion of “weakly” Bloch and Neel skyrmions. What would be the helicity value (the range of helicity values) to differentiate between different spin structures. In Fig. 7 (supplement), the authors show some helicity values, but again somehow hide the information, since the color corresponding to Bloch, Neel, Phase II and III skyrmions is not reflected in the figure.

4. The authors always refer to skyrmions as multi- q states obtained by intersection of spirals. I wonder what would be the structure of isolated skyrmions according to the functional (1)? Is the helicity of isolated skyrmions somehow fixed? I believe it is straightforward to obtain solutions for isolated skyrmions. So, there must be no problem to address this question. I thank you in advance.

5. It is not quite clear. Is EuNiGe_3 an antiferromagnet? At least, other papers on the subject claim so. Then, the impact of the magnetic field must be quite different than in a ferromagnet? Also, different phases would be depicted differently.

A. I would also ask the authors to check other papers on the same compound, which have been omitted. For example, arxiv: 2306.14767 or 2306.12669 or J. Phys. Chem. Lett. 2023, 14, 4, 1000-1006.

6. The authors performed lots of different experiments. But somehow all these measurements are not connected by any clear message.

7. I would appreciate to have more details on the theoretical model. Indeed, the model appears at the end of the paper in Eq. (3). It includes many parameters, the role of which is not explained. Then, the authors just ascribe some values to those parameters and obtain different spin structures. I would appreciate some justification at least in the supplement.

8. The related question: from the theoretical model, it is quite hard to judge what is the stabilization mechanism of skyrmion phases. It can be DMI, it can be higher-order exchange interaction, it can be RKKY ... In this case, it is not a big surprise that the phases with different helicities are stabilized.

9. I wonder if the authors minimized the obtained spin structures with respect to the period? Would it be possible to show that skyrmion phases correspond to the minima of the energy functional and are not some confined states distorted by the boundary conditions?

In conclusion. Although the manuscript contains a lot of different experimental and theoretical results it is very hard to understand the main message. I would really appreciate if the authors conveyed their idea clearly and would not hide very important information. Indeed, the paper is devoted to the change of skyrmion helicity, but the resolution of figures is so low that it is impossible to resolve anything.

Reviewer #3 (Remarks to the Author):

In this manuscript, Singh et al. reported the observation of magnetic skyrmion phase transitions in a polar tetragonal magnet EuNiGe_3 . Small angle neutron scattering was used to map the magnetic phase transition across cryogenic temperatures and external magnetic fields, where two out of the four phases were found to exhibit skyrmions with mixed helicities. The phenomenon observed in this manuscript is interesting for the community. Therefore, I recommend publish this work in Nature Communications after the following comments were addressed by the authors.

1. Fig. 1e is mislabeled as 1d.

2. In Figs. 1a-b, 1e-f, the authors showed the schematics of the skyrmions with various helicities. However, it is not clear to me from the schematics how the spin arrangements are different in mixed helicities (Figs. 1e-f) from the fixed helicities (Figs. 1a-b). Since it is one of the main findings in this work, it would be beneficial to show clearly how the spin arrangements are different from the pure Bloch- or Neel-characters. (e.g., different color schemes for highlighting the spin orientations, higher pixel resolutions, etc).

3. Strictly speaking, the first term in eq.2 should be vorticity, as defined in previous literatures [1]. This vorticity will be equivalent to skyrmion number N_{sk} , only if the spin at the periphery is anti-parallel to that at the core. However, in the current manuscript, it is not clear to me that there is evidence of ruling out the possibilities for merons, which have spins confined in-plane at the periphery.

4. At pg.5, line 81, the author claimed the “microscopic evidence for a hybrid helicity of skyrmions as an intrinsic property in bulk magnets has remained elusive.” However, skyrmions with mixed helicities have been reported more than 10 years ago [2].

5. At pg.5, line 89, it would be beneficial for the readers to specify each of the Phases 1-4.

6. In Figs. 3e-h, the authors justify the ellipticity of SANS spots as elongation of skyrmions along various Q directions. However, in Fig. 3f, 3g & 3h, it appears to me that there are split spots. Please explain this feature.

7. In Fig. 3h, there are weak reflections in regions from $(q_x, q_y)=(-2, -2)$ to $(q_x, q_y)=(2, 0)$. Please explain such features.

8. From symmetry perspective, the non-centrosymmetric crystal structure of EuNiGe_3 allows for the DMI to occur, which usually leads to skyrmions with same helicity. Using spin DFT and Monte Carlo simulations, the authors explained the hybrid helicity skyrmions originated from the frustration caused by Q-dependent DMI or anisotropic exchange interactions. If such frustration exists, why do the skyrmions still form lattices instead of aperiodic structures?

References:

1. N. Nagaosa and Y. Tokura, *Nature Nanotechnology* 8, 899 (2013), “Topological properties and dynamics of magnetic skyrmions”.
2. X. Yu et al., *PNAS* 109, 8856 (2012), “Magnetic stripes and skyrmions with helicity reversals”.

Reviewer #1 (Remarks to the Author):

Reviewer: D. Singh et al. investigated the magnetic field dependent magnetic properties of an itinerant polar magnet EuNiGe₃, via a combination of experimental techniques, including magnetization, magneto-transport and neutron scattering. A magnetic field induced transition between hexagonal and square crystals of hybrid skyrmions is observed. A theory that invokes Heisenberg exchange and Dzyaloshinskii-Moriya interactions (DMI) is proposed to explain the experimental findings.

Reviewer: Although some experimental works had been done before in EuNiGe₃ using similar techniques, this study presents a more complete investigation of its magnetic field dependent properties. Particularly, the magnetic field induced transition between hexagonal and squares crystals of magnetic skyrmions is interesting. Also, the theoretical reasoning for this observation can be applicable to a broad class of magnetic materials. Considering the significance, this work has the potential of being accepted for publication at Nature Communications. That being said, some concerns need to be addressed before the publication of this manuscript.

Response: We thank the Reviewer for their time carefully reading the manuscript and for their positive assessment. In the following, we respond to the Reviewer comments and describe changes applied to the manuscript.

Reviewer: 1) What are the valence states of Ni and Ge atoms in EuNiGe₃? Do they also have magnetic moments that may alter the picture presented in this work?

Response: EuNiGe₃ is an intermetallic compound with an established metallic character over the examined temperature and field range, indicating the delocalization of valence electrons from constituent atoms. Thus, concerning the valences, conventional ionic and covalent bonding concepts like Lewis-Kossel or Langmuir models do not apply. It is well-known that intermetallic compounds host the so-called multicenter-bonding scheme, making it challenging to apply traditional valence counting rules. Detailed electronic structure analysis can be in principle achieved through spectroscopic experiments like XPS and ARPES, though such data remain missing as far as we are aware. Single crystal diffraction could provide insights into spatial charge redistribution, but this lies beyond the present scope. Therefore, we hesitate to conclude definitively on Ni and Ge valence states at this stage.

On the more important aspect of the origin of the magnetic moments, according to recent X-ray magnetic circular dichroism (XMCD) measurements [K. Chen *et al.*, J. Phys. Chem. Lett. **14**, 1000 (2023)], a negligible XMCD intensity is observed at the Ni L-edge, suggesting the Ni in EuNiGe₃ is non-magnetic. As far as we are aware, a microscopic measurement for Ge spin polarization does not exist. We thus performed a SDFT calculation which suggests moments due to either of Ni and Ge are negligibly small (less than 1% of Eu). There thus appears to be little doubt that Eu alone governs the magnetic properties of EuNiGe₃. To make this point, we updated the description of the magnetic properties on lines 116-118, citing the XMCD study as reference 31 in the main text.

Reviewer: 2) Magnetic field dependent evolution of the propagation vectors is measured via the small angle neutron scattering (SANS) technique. However, it is not clear whether there is any non-

zero contribution in the z-component of these vectors, especially the case when magnetic field is turned on in phase regions II-III. This concern need to be discussed.

Response: In none of the magnetic phases did we observe a finite z-component in the SANS range (or L-component in reciprocal lattice units) for any of the magnetic modulations. This information is available on pages 9 through 11 in terms of the zero value of the L-component of the propagation vectors Q_1 to Q_7 . To make this point clearer, we added extra information to caption of Figure 3 on lines 773 to 774.

Reviewer: 3) Lines 32-34: the authors present a theory that invokes multiple exchange and DMIs to explain the field induced transition between hexagonal and square skyrmion lattices. But the key information in the manuscript is not reflected from this sentence. Heisenberg exchange couplings and DMIs are commonly present in lots of magnetic materials, either insulating or metallic.

Firstly, “multiple exchange” is a very vague term to describe Heisenberg exchange couplings, either isotropic or anisotropic, from different origins.

Response: We thank the Reviewer for this comment. To make the formulation clearer, in the revised abstract we changed “multiple” to “momentum-resolved” (line 33) to describe that the interactions supporting the different phases compete in momentum space at the observed magnetic wavevectors of those phases. The change is made similarly on line 108 too.

Reviewer: Secondly, the authors argue that the generalized RKKY interaction is important for the stabilization of the observed skyrmion lattices. In addition, the authors argue that the low symmetry Q vectors are important for the observation of magnetic field induced transition between hexagonal and square skyrmion lattices. It seems that these key ingredients are not specified in the abstract.

Response: We agree with the Reviewer that both factors are important. RKKY interactions and the low symmetry of the Q-vectors are now included in the revised abstract and described as key ingredients for the theoretical modelling that allow us to understand the stabilization of the observed skyrmion phases.

Reviewer: Lines 272-273: On a related note, the authors need to be specific about the different parameters (αQ , βQ , γQ) presented in the Heisenberg exchange model and discuss the origins for these terms. For instance, does spin-orbit coupling play a role here?

Response: The origin of the mentioned parameters comes from the RKKY interaction (Fermi surface instability) and indeed from the spin-orbit coupling as well. Therefore, in general, these parameters depend on the electronic band structure in the system, as discussed in references 17 and 50. We have added this important information to the main text on lines 294-296.

Reviewer: 4) The experimental resolution of SANS and polarized SANS need to be mentioned. The corresponding errors of the experimental data need to be included as well.

Response: We have updated the Small-angle neutron scattering (SANS) measurements subsection in the Methods to provide estimates of both wavelength and momentum-space resolution for the unpolarized and polarized SANS measurements (lines 440-443 and 462-466). In the captions of Figures 3 and 4, Table 1, and Supplementary Figures 3, 4, 5 and 6, we have added a statement on the corresponding meaning of the uncertainties of data points obtained from SANS data analysis. We also re-checked the error propagation throughout the manuscript and consequently we have revised

Supplementary Figures 3, 4, 5, 6 to display improved indications of measurement uncertainty. Where error bars are not visible, it is declared in the caption that they are smaller than the symbol size.

Reviewer: Some additional comments for this manuscript:

Reviewer: 1) Some figures presented in the current version of this manuscript are of very low quality. Also, the font size in some figures should be larger and consistent with the other figures (such as Fig 1d and fig 6). Particularly, figs. 6a-h are too small compared to other figures, such as fig. 5.

Response: Following the Reviewer's comment, we have revised Figures 1, 2, 3, 4 and 6. For Figure 1, we have improved the resolution of Figure 1d, and aimed at font standardization in this figure. We also improved the resolution of Figures 3 and 4. For Figure 6, we agree that Figures 6a-h appeared too small, and thus we have revised the figure so that the panels are larger on the page. With the improved Figure resolution, the spin textures can be studied in better detail on zooming in. To avoid a cumbersome layout of Figure 6, the former Figure 6i has become a new Figure 7.

Concerning the standardization of font sizes across *all* figures, we judge that the font size is nearly uniform. We conclude this, also when considering that the figures may have different sizes in a final article compared with the review draft submitted in Microsoft Word format. An exception is main text Figure 2 where the font-size was proportionally larger compared with the other figures. Therefore, we revised Figure 2 to have a font size closer to that of the other figures.

Finally, although not explicitly mentioned by the Reviewer, we also revised the colourmaps encoding the out-of-plane magnetization of the Phase II and Phase III skyrmion lattice schematics in Figures 1g and h. Previously the colourmap used for these sub-figures was the same that used for encoding the helicities relevant for Figures 1a, b, e, and f. We considered that using the same colourmap twice in the same figure for different parameters could be a potential source of confusion, which the implementation of a colourmap helps us to avoid.

Reviewer: 2) Lines 46-52: some symbols introduced here are not well defined. Some general readers may find them confusing.

Response: Motivated by this comment, and a similar one from the 3rd Reviewer (their comment 3), we have revised the relevant part of the introduction on lines 46 -56 to be both clearer and more rigorous in definitions of parameters.

Reviewer: 3) Line 171: how is that a 90 deg rotation?

Response: When the magnetic order breaks the symmetry of the host crystal, multiple magnetic domains appear that can be mutually transformed into one another by the broken symmetries. Phase II indeed breaks the fourfold symmetry of the host crystal such that we observe at least two magnetic domains which are related to one another by a fourfold rotation, i.e. a 90° rotation around the polar axis. This can be seen visually in Fig. 3f and j, whereby one hexagonal domain can be transformed into the other by a simple 90° rotation of the triple Q-vector distribution around the polar axis. We changed the text on lines 186 to 188 to clarify the relationship between the magnetic domains and broken fourfold crystal symmetry in Phase II.

Reviewer: 4) Line 207-215: for the polarized SANS, how will the non-perfect spin-flip and non-spin-flip affect its ratio?

Response: If imperfections in polarizing efficiency of the setup would not be taken into account, this would indeed impact the ratios of non-spin-flip (NSF) and spin-flip (SF) SANS intensities reported in Table 1. In our experiment and analysis however, we followed standard measurement procedures so that we could apply corrections that eliminate the effect of imperfect polarizing efficiency on the measured NSF and SF SANS intensities. To make this point clearer, in the Small-angle neutron scattering (SANS) measurements subsection in Methods, we updated the relevant sentences on lines 467 to 469.

Reviewer: 5) Line 325-338: it appears to me that there exists some competition rather than “frustration”.

Response: We appreciate the Reviewer pointing this out and agree with their supposition. We changed all three instances of ‘frustration’ to ‘competition’ on lines 350, 352 and 360.

Reviewer: 6) Line 379, powder x-ray diffraction is not a direct method to confirm the sample purity. Citing an early work or presenting with more data will help.

Response: We thank the Reviewer for this comment. In response we performed single crystal x-ray diffraction on our crystals, and with separate refinement runs we confirmed the occupancies of all atoms were always at the nominal composition values within uncertainty. Our refinements further confirmed that our crystals to display the crystallographic parameters previously reported in references 32-34. Finally, since the characteristic temperature- and magnetic-field dependent phase transitions, such as the Néel temperature, $T_N = 13.2$ K, are in good agreement with previous studies reported in references 32–35, we suggest our EuNiGe_3 single crystals have purities that are comparable to crystals studied in earlier work. Accordingly, on lines 403-416 we improved our description of crystal quality with respect to both the new X-ray diffraction measurements done on our crystals, and the previous studies of this system.

Reviewer: 7) Line 670, Fig 2. χ_{ac} was used to describe the ac susceptibility. This should be applied to the main text as well.

Response: We thank the Reviewer for their careful reading. We corrected the main text so that mentions of the ac magnetic susceptibility χ_{ac} include the ‘ac’ subscript (on lines 134, 136, 138, 139, 755). We likewise updated the text similarly in the Supplemental Information on lines 26, 27, 32, the Supplementary Fig. 1 caption, and the axis labels of Supplementary Figure 1a and b.

Reviewer: 8) Fig 3 and table I: multiple magnetic Q vectors are defined when describing its magnetic field dependence. Some readers may get confused. The authors may consider updating the caption and the table with more details, such as $\mathbf{Q}_1 \equiv \mathbf{Q}_7$ and $\mathbf{Q}_7 = \mathbf{Q}_5 + \mathbf{Q}_6$.

Response: Thank you for the suggestion aiming at improving the clarity of our work. Due to the existence of multiple magnetic Q-vectors amongst the different phases, we aimed to be a succinct and clear as possible in their presentation, particularly in Figure 3. This is the main motivation for the panels i to l, which shows clear definitions of these Q-vectors (and equivalents). Concerning Figure 3, we prefer to not write $\mathbf{Q}_1 \equiv \mathbf{Q}_7$, because we interpret \mathbf{Q}_1 as a fundamental modulation vector of phase I, and \mathbf{Q}_7 a higher-order modulation vector of phase III. So, these two modulation vectors are not

equivalent, even if they are located at very similar positions in momentum space. Likewise, even \mathbf{Q}_1 in phase I and \mathbf{Q}_8 in phase IV cannot be written as equivalent, even if they appear rather similar. On the other hand, we agree that we can state $\mathbf{Q}_7 = \mathbf{Q}_5 + \mathbf{Q}_6$, and have added this information to the Fig. 3 caption (lines 778-779). In Table I, the second column defines Q -vector *type* according to the definitions in Figs. 3, and as such it is inappropriate to define relations between different Q -vectors here. We nonetheless updated the Table 1 caption on line 824-825 to make clear the definitions of the different Q -vectors.

Reviewer: 9) Fig 6 and Supplementary Note VII: the authors may consider adding a table including all the parameters used for the simulations in Fig 6 under different magnetic fields.

Response: Since all the parameters used for the simulations are already listed in Supplementary Note VII, we prefer to leave this information in the Supplement, rather than adding a new table to the main text. On line 315, we updated the text to make clear that the simulation parameters can be found in Supplemental Note VII. There it is also clear that we use a common set of model parameters for the simulations (with solutions found for each phase shown in Figure 6). Only the magnetic field H^z is varied in the simulation.

Reviewer: To summarize, the experimental observations and theoretical arguments in this work are interesting and can have a broad impact to the community focusing on magnetic skyrmions. But some concerns need to be addressed and some key information need to be highlighted before the acceptance for publication.

Response: We thank the Reviewer for their assessment that our work may have a broad impact on the magnetic skyrmion community. We hope that the Reviewer finds that we have satisfactorily responded to their concerns and may consider the revised manuscript as suitable for publication in Nature Communications.

Reviewer #2 (Remarks to the Author):

Reviewer: In the present manuscript, the authors investigate a direct transition between hexagonal and square crystals of hybrid skyrmions in a polar magnet EuNiGe₃. To prove the concept, the authors utilize versatile experimental techniques such as bulk magnetic and transport measurements, measurements of the Hall resistivity, SANS, susceptibility measurements etc. The authors convincingly prove that the ultra-small skyrmions change their helicity from Bloch to Neel fashion by an applied magnetic field.

Reviewer: The paper is certainly of great use for skyrmionics and can revolutionize various applications of skyrmions in particular involving a dynamic reconfiguration of skyrmion-based magnonic crystals. Moreover, the paper is written by a pioneer of skyrmionics Prof. Y. Tokura who discovered skyrmions and introduced the modern language of skyrmionics. In particular, in Ref. [5], Prof. Tokura introduced the notion of the topological charge, which allowed to classify skyrmions and anti-skyrmions. The manuscript is well-written and clear and I recommend it to publication. In the following, I just would like to raise some points, which could be amended.

Response: We thank the Reviewer for their time reviewing our manuscript. We were pleased to read that the Reviewer believes “The manuscript is well-written and clear and I recommend it to publication”. In the following, we respond to the constructive comments of the Reviewer, which are helpful for making our work more persuasive.

Reviewer: 1. The motivation of the manuscript is not quite clear to me. Is it a phase transition between hexagonal and square skyrmion lattices or the change of skyrmion helicity during this phase transition? Unfortunately, after reading the manuscript, I cannot clearly express what is the main idea of the paper?

Response: We appreciate this question regarding the motivation of our work. The magnetic field-driven phase transition between skyrmion Phases II and III involves significant, concomitant changes in both skyrmion helicity and skyrmion crystal coordination. While both aspects contribute to our work's motivation, we can agree that this was not conveyed as well as possible in key areas of our original submission. In response, we have now revised the manuscript's title and abstract (lines 28-35) to emphasize the importance of both effects. Overall, and considering the entire content of the paper, we now prioritize more the observed helicity change, which we believe to be a genuinely new observation for the bulk of a skyrmion hosting system, alongside the skyrmion crystal structure change, which has been previously observed albeit in different contexts (see next comment).

Reviewer: A. If the motivation of the paper is the square or the hexagonal alignment of skyrmions, then, one can recall some papers, in particular [T. Nakajima, H. Oike, A. Kikkawa, E. P. Gilbert, N. Booth, K. Kakurai, Y. Taguchi, Y. Tokura, F. Kagawa, and T. Arima, Skyrmion lattice structural transition in MnSi, *Sci. Adv.* 3, e1602562 (2017)]. The authors observed the transition from the hexagonal to the square skyrmion order. In this sense, there are many papers with the same motivation.

Response: The key difference between the quoted study on MnSi and our work, is that the skyrmion coordination transition observed in MnSi is only observed in an out-of-equilibrium (metastable) condition. On the other hand, in the present work we report the hexagonal-to-square skyrmion crystal transition to take place between two thermodynamic equilibrium phases that are adjacent to

each other in the phase diagram. As far as we are aware, this is the first time such a skyrmion crystal transition has been reported between equilibrium skyrmion phases in a bulk material. To emphasize the distinction between our work and the study of the far-from-equilibrium skyrmion crystal coordination transition in MnSi, we modified the text at lines 380-382, and cite the above-mentioned reference as reference 52. We also cite reference 53 as another example of a bulk observation of a skyrmion coordination transition in a far-from-equilibrium condition.

Reviewer: B. If the motivation of the paper is the value or change of the skyrmion helicity between 0 and $\pi/2$, which differs from the Bloch or Neel fashion of rotation, then, one would first recall the chiral magnets with C_n symmetry. In particular, in Ref. [Chem. Rev. 2021, 121, 5, 2857–2897], Prof. Tokura explicitly introduced the Lifshitz invariants for this point-group symmetry in Eq. (9). So, I wonder why this symmetry class is not even mentioned in the present manuscript?

Response: This is a good point from the Reviewer. While in the introduction we introduced the general expectation for the skyrmion-type according to the point symmetry of the host crystal, we did not explicitly state the C_{4v} point symmetry of EuNiGe_3 . We now do so on lines 94 and in the Figure 1 caption on line 737.

Reviewer: C. Then, one could recall frustrated skyrmions, which change their helicity or ultra-small skyrmions in monolayers with induced DMI etc.

Response: While the helicity variation of frustration-stabilized skyrmions has been considered theoretically, we could not find experimental evidence, despite an extensive literature search. If we have missed such a study, we are glad to be informed by the Reviewer. We can nonetheless agree that it is important to highlight other suggestions for, and examples of, settings for skyrmions with non-standard helicities. On lines 85-86 we now include references 25 and 26 that mention important theoretical studies of frustrated skyrmions, and reference 28 as a relevant example of experimental research of helicity control in ultra-small skyrmions in monolayers with induced DMI.

Reviewer: D. One can also recall the change of skyrmion helicity near the surfaces of thin-film helimagnets, the so-called surface twists.

Response: Similarly, as for the previous comment, we now include references 29 and 30 on line 86 as a relevant example of modulated skyrmion helicities at the surfaces of helimagnets.

Reviewer: 2. I really apologize for my complaint but the resolution of figures 1 and 6 is so low that it is impossible to discern the structure of different spin distributions. Indeed, the authors speculate that the helicity of skyrmions changes, but eventually it is impossible to see any spin structure, the information is hidden. I wonder why?

Response: We thank the Reviewer for this comment. We agree that amongst the originally submitted figures, it was difficult to discern the detailed magnetisation textures amongst the different phases. To address this concern, we have firstly improved the resolutions of Figures 1 and 6 (and revised the layout of Figure 6) such that on zooming in the detailed spin distributions can be examined more clearly. We also changed the layout of Figure 6 so that the simulated spin structures are also more visible on zooming in. To accommodate the new layout of Figure 6, the old Figure 6i has become a new Figure 7. For Figures 1g and h, we also changed the colourmap encoding the out-of-plane magnetization for skyrmion lattice Phases II and III. Previously this colourmap was the same as that used to encode the spatial distribution of the skyrmion helicity in Figures 1a, b, e, f. We

applied the change in colourmap to avoid any potential confusion between calculated skyrmion helicities and out-of-plane magnetization that may have arisen in the original version of the figure.

Concerning our conclusion that the skyrmion helicity changes between Phases II and III, Figure 1 is prepared so that the helicity distribution of the skyrmion encoded by the relevant colourmap can be seen from inspection of panels e and f. As written at the start of the discussion, these helicity values are calculated according to the magnetisation textures expected according to the results of the Monte Carlo simulations reported in Figure 6. Since the resolutions of both Figure 1 and 6 are now improved, as well as the layout of Figure 6 also improved, the associated spin textures can now be inspected more easily.

Nonetheless, we agree with the Reviewer that it is important to provide improved visualisations of the magnetisation textures to clarify their connection to the helicity. Firstly, we revised the spin texture schematic components of Figures 1a, b, e and f so that the textures for the various skyrmion types are much clearer than before. Secondly, since the spin textures shown in Figs 1e and f are respectively generated from the results shown in Figures 6b and c, we hesitate to introduce a third visualization and detailed spin texture analysis into the main text. Therefore, we significantly extended Supplementary Note IX with further text and a new Supplementary Figure 8 to clarify further the connection between the spatial distributions of the magnetisation textures and helicities for each skyrmion type. With these additions, we provide more transparent evidence for how the helicity distributions determined in Phases II and III contrast not only with each other, but also with those expected for standard Bloch- and Néel-type skyrmions.

The overall distribution of helicity values across the skyrmion lattice unit cell remain considered quantitatively in Supplementary Note IX, with the detailed distributions for each skyrmion type visualized more clearly in a revised Supplementary Figure 7, along with updated accompanying text.

Thus, the detailed visualisation of the different skyrmion spin structures depicted in main text Figures 1e, 1f, 6b and 6c are now shown and discussed in improved detail both in the main text and Supplementary Information. We hope that with these changes, the Reviewer may be satisfied that details of the magnetisation textures are not hidden and that the connection between them and the spatial dependence of the skyrmion helicity is clearer than before.

Reviewer: A. I would recommend to show the structure of all phases. I believe that one image would immediately clarify the point raised. In particular, is it possible to see the structure of a spiral state with the wave vector rotating in-plane perpendicular to the field?

Response: With the new high-resolution version of Figure 6, panels a-d indeed provide real-space visualisations of the magnetic structures in all Phases I to IV according to the simulation results, which are consistent with the experimental data at hand. We believe that the visualisations of the spin textures in Figure 6, Figure 1, and the new Supplementary Figure 8 in Supplementary Note IX, provide sufficiently detailed and transparent views of the proposed magnetic structures in real-space.

Concerning the second part of the comment, with respect, it is not clear why we may consider presenting an image of a “spiral state with the wave vector rotating in-plane perpendicular to the field”. In none of the observed Phases I to IV does the wave-vector direction rotate in such a way. As reported in the manuscript and Supplementary Note IV, experimentally we find the wave-vector

directions to always remain rigidly in the tetragonal plane, for fields applied either along the c- or a-axes. For the latter geometry, there is no observed reorientation of the magnetic wave vector with increasing field, indicating that for the direction of applied field, the spiral state likely just acquires a conical-like distortion.

Reviewer: B. Then, I would recommend to describe in more detail the method how the helicity was defined. If I understand correctly, for very small skyrmions when the angle between adjacent spins is large (BTW, what is the angle?), one should first determine the skyrmion center. After that, one should measure the helicity along the radial direction. Does the helicity changes along the radial coordinate or it remains constant?

Response: We agree with the Reviewer that for ultra-small skyrmions like those considered in EuNiGe_3 , the angle between adjacent spins may be described as large (in comparison say to skyrmions in chiral magnets). Since the present model simulations and experimental results do not provide an atomistic description of the magnetic structure, we cannot comment on the mutual angles between specific spins.

Instead, the simulations provide insight on the spatial distribution of the magnetization over the experimentally relevant magnetic length scales, which for clarity we depict as a spin structure on a square grid lattice. Subsequently, the skyrmion center is determined according to the location of the maximum of the out-of-plane magnetization, which itself can be tied to a grid site since the global phase of any incommensurate magnetic structure can be freely defined. To make clear how the magnetisation textures are treated in real-space, we extended Supplementary Note IX with lines 273-299. Subsequently, as described in the Figure 1 caption, the helicity is evaluated with respect to the center of the skyrmion using the expression $\chi = \arccos(\hat{\mathbf{n}}_{\perp} \cdot \hat{\mathbf{r}})$. Here, \mathbf{n}_{\perp} is the normalised moment component perpendicular to the out-of-plane direction \mathbf{c} , given by $\mathbf{n}_{\perp} = \mathbf{n} - \hat{\mathbf{c}}(\hat{\mathbf{c}} \cdot \mathbf{n})$. We have updated the caption in Figure 1 to describe that parameter $\hat{\mathbf{r}}$ is the position relative to the skyrmion core (main text, line 734).

Finally, as can be seen in Figures 1e and f, Supplementary Figure 8, and as discussed in more detail in the updated Supplementary Note IX (lines 338-377), it is seen that the helicity of the skyrmions in Phases II and III may indeed vary with $\hat{\mathbf{r}}$ along both the radial direction from the skyrmion core, and azimuthal angle within the skyrmion lattice plane.

Reviewer: C. It can be the case that for such small skyrmions and periodic boundary conditions applied, one may numerically obtain different solutions. For example, in one solution, a skyrmion center may be located at the grid knot, in another – between the points of the numerical grid. I wonder, how the authors control this process? Could the authors show their solutions or this information is also hidden?

Response: In the numerical simulations, the spin configuration was always optimized based on simulated annealing. There, it was always found that the skyrmion core (center) was located at an interstitial position in the square numerical grid. Thus, the energy for the state where the skyrmion core is located at the interstitial position is lower than that for the state where the skyrmion core is located at the lattice site. Concerning the request to show solutions, we would respectfully point out that the spin textures presented in Figure 6a-d are the solutions of the Monte Carlo simulations,

with these solutions confirmed to be the lowest energy states. Thus, from the perspective of the simulations performed in the manuscript, no information is hidden.

Reviewer: 3. A related question: the authors introduce the notion of “weakly” Bloch and Néel skyrmions. What would be the helicity value (the range of helicity values) to differentiate between different spin structures. In Fig. 7 (supplement), the authors show some helicity values, but again somehow hide the information, since the color corresponding to Bloch, Néel, Phase II and III skyrmions is not reflected in the figure.

Response: Following the Reviewer’s concern, we revised the layout of Supplemental Figure 7, in order that the information contained within it is presented more clearly. Now the distribution of helicity values associated with the different skyrmion types can be more easily inspected and contrasted. It becomes clear that the skyrmions we discover in Phases II and III fit neither the standard Bloch nor Néel skyrmion expectation.

To put the notion of “weakly” Bloch-like and “weakly” Néel-like skyrmions on firmer ground, in the revised Supplementary Fig. 7 we now include the measures of the average helicity $\bar{\chi}$ and standard deviation σ for each histogram. We find for Phase II that $\bar{\chi} = 1.428$ with $\sigma = 0.105$, and so the spread of helicities is mainly less than the value of $\pi/2$ expected for the pure Bloch skyrmion lattice. In the absence of any formal terminology, we invoked the term “weakly” Bloch-type to describe Phase II, since it is clearly not pure Bloch, yet obviously closer to Bloch than Néel. For Phase III, we find that $\bar{\chi} = 0.581$ with $\sigma = 0.223$, which is closer to Néel than Bloch and so we describe the Phase III skyrmions as “weakly” Néel. One can suppose that the switch between allocation between “weak” Bloch and “weak” Néel occurs at a value of $\bar{\chi} = \pi/4$, where no overall helicity type is predominant. We have updated and extended Supplementary Note IX (lines 301-336), with these considerations. Ultimately, the terms “weakly” Bloch and “weakly” Néel indicate an overall predominance of either the Bloch or Néel type winding of the hybrid skyrmions in Phases II and III. We hope that the Reviewer may find the improvements acceptable in the absence of an established nomenclature for the hybrid skyrmion helicity values we observe.

Reviewer: 4. The authors always refer to skyrmions as multi-q states obtained by intersection of spirals. I wonder what would be the structure of isolated skyrmions according to the functional (1)? Is the helicity of isolated skyrmions somehow fixed? I believe it is straightforward to obtain solutions for isolated skyrmions. So, there must be no problem to address this question. I thank you in advance.

Response: For the model discussed in the manuscript, it is difficult to obtain the solution of isolated skyrmions, since this model has interactions only at specific wave-vector channels. Meanwhile, we can speculate that the structure of isolated skyrmions might be similar to that of periodic skyrmions as shown in Fig. 6, where the helicity is fixed at values that are neither 0 nor $\pi/2$. Solutions for isolated skyrmions cannot be obtained in the present model since it only includes long-range interactions in momentum space. We would like to mention that in the present work, the simulations aim at supporting the interpretation of our observations of ordered skyrmion crystals, rather than isolated skyrmions for which we have no experimental evidence. Therefore, while we certainly believe that seeking an appropriate theoretical description for isolated skyrmions in this material is an interesting topic, we feel that its inclusion in the present manuscript respectfully lies beyond the current scope.

Reviewer: 5. It is not quite clear. Is EuNiGe₃ an antiferromagnet? At least, other papers on the subject claim so. Then, the impact of the magnetic field must be quite different than in a ferromagnet? Also, different phases would be depicted differently.

Response: Experimentally EuNiGe₃ displays magnetic structures with finite incommensurate wave vectors. This implies that this material belongs to the class of spiral magnets different from conventional antiferromagnets. As discussed in the main text, our theoretical analysis suggests that the origin of the spiral magnetism arises from the RKKY interaction characteristic of itinerant magnets. Consequently, in EuNiGe₃, the magnetic field allows us to tune the delicate competition between different interactions competing at different wave vector instabilities defined by the Fermi surface nesting. This behavior clearly contrasts to that of a normal ferromagnet too.

Reviewer: A. I would also ask the authors to check other papers on the same compound, which have been omitted. For example, arxiv: 2306.14767 or 2306.12669 or J. Phys. Chem. Lett. 2023, 14, 4, 1000-1006.

Response: Thank you for bringing these references to our attention. The first preprint describes a recent resonant x-ray scattering study of this material, wherein evidence for Phase II is presented. In contrast to our work however, Phase III appears to be missed in that study, and no theoretical analysis of the Phase II stability or internal texture is given. Nonetheless, it is appropriate to mention the first arxiv Reference, and we do so as Reference 36 in the text added on lines 127-130. Concerning the second arxiv paper, this reports a study of an isostructural but different material EuIrGe₃. Since this material seemingly is not found to host skyrmions and displays spiral states different to those we observe in EuNiGe₃, we respectfully decide to not make mention to this work. The third paper concerns high pressure synchrotron x-ray measurements of EuNiGe₃. In this paper lies valuable information concerning the magnetic properties of the material, and we now include this paper as Reference 31 on line 118.

Reviewer: 6. The authors performed lots of different experiments. But somehow all these measurements are not connected by any clear message.

Response: Thank you for highlighting your concern, which seems similar to the first comment questioning the motivation for the work. As explained above, we revised both the title and abstract to transmit the main messages of the manuscript more clearly, and more in-line with the main body of the manuscript. We hope that the Reviewer now better understands the main message of our work, which we intend to be the observation of the hybrid skyrmion textures in EuNiGe₃, and the observed modification of these hybrid textures concomitant with the transition in skyrmion crystal coordination at the Phase II to Phase III boundary.

Reviewer: 7. I would appreciate to have more details on the theoretical model. Indeed, the model appears at the end of the paper in Eq. (3). It includes many parameters, the role of which is not explained. Then, the authors just ascribe some values to those parameters and obtain different spin structures. I would appreciate some justification at least in the supplement.

Response: We thank the Reviewer for the comment, which aims at enhancing the transparency of our manuscript. Following the Reviewer's advice, we have updated Supplementary Note VII to explain the roles of the various parameters used in the simulations on lines 230-240.

Reviewer: 8. The related question: from the theoretical model, it is quite hard to judge what is the stabilization mechanism of skyrmion phases. It can be DMI, it can be higher-order exchange

interaction, it can be RKKY ... In this case, it is not a big surprise that the phases with different helicities are stabilized.

Response: We thank the Reviewer for this important comment. From our simulations, we can suggest that the stabilization mechanisms of the two skyrmion phases are different. The hexagonal skyrmion Phase II is stabilized by the interplay between the easy-axis interaction $\alpha_{\mathbf{Q}_v}$ and symmetric anisotropic exchange interactions $\beta_{\mathbf{Q}_v}$ and $\gamma_{\mathbf{Q}_v}$. On the other hand, the square skyrmion phase (Phase III) is stabilized by the DMI $\mathbf{D}_{\mathbf{Q}_v}$. This information is now added to the main text on lines 329-333.

Reviewer: 9. I wonder if the authors minimized the obtained spin structures with respect to the period? Would it be possible to show that skyrmion phases correspond to the minima of the energy functional and are not some confined states distorted by the boundary conditions?

Response: In the present model, we considered the momentum-resolved interaction at specific wave vectors. In this sense, the periodicity of the skyrmion lattice is determined by the interaction. In other words, skyrmion phases correspond to the minima of the energy functional with respect to the period, because the periodicity of the skyrmion lattice is assumed. For example, when considering the skyrmion lattice with the ordering wave vectors deviated from Q2, Q3, Q4, i.e., Q2 + delta, Q3 + delta, Q4 + delta in Phase II, the energy of the skyrmion lattice is zero, since we do not consider the interaction at Q2 + delta, Q3 + delta, and Q4 + delta. Thus, even when changing the boundary conditions, we always obtain the skyrmion solution with Q2, Q3, and Q4 within the present model.

Reviewer: In conclusion. Although the manuscript contains a lot of different experimental and theoretical results it is very hard to understand the main message. I would really appreciate if the authors conveyed their idea clearly and would not hide very important information. Indeed, the paper is devoted to the change of skyrmion helicity, but the resolution of figures is so low that it is impossible to resolve anything.

Response: We thank the Reviewer for raising their concerns and providing constructive advice in their comments on how these concerns may be alleviated. Motivated by their comments, we have reformulated the key areas of the title and abstract to help convey the message of our manuscript more clearly. This way, these key areas of the manuscript are now aligned with what the Reviewer writes above as the most prominent finding, which is that the work is “devoted to the change of skyrmion helicity”. We would therefore like to thank the Reviewer for their advice which helped us to refine the message of our work.

We were also concerned by the Reviewer’s suggestion that we, the authors, hid very important information. According to their report, the Reviewer wrote in three comments where they found information to appear as hidden, namely comments 2, 2C, and 3. We take these comments seriously and as described in our detailed replies, we have applied changes (where relevant) aiming at full transparency in the mentioned areas. Not only do we now provide additional supporting material that show the connection between the magnetisation textures and helicity of the skyrmion phases in much more detail, but we also revised most of the figures with improved resolution. We believe these changes have led to an improvement in the overall quality of the manuscript.

Having addressed each of the Reviewer comments one by one, we hope the Reviewer may be satisfied with the revised manuscript and find it acceptable for publication in Nature Communications.

Reviewer #3 (Remarks to the Author):

Reviewer: In this manuscript, Singh et al. reported the observation of magnetic skyrmion phase transitions in a polar tetragonal magnet EuNiGe₃. Small angle neutron scattering was used to map the magnetic phase transition across cryogenic temperatures and external magnetic fields, where two out of the four phases were found to exhibit skyrmions with mixed helicities. The phenomenon observed in this manuscript is interesting for the community. Therefore, I recommend publish this work in Nature Communications after the following comments were addressed by the authors.

Response: We thank the Reviewer for their concise evaluation and positive assessment. We are pleased to read that they think our work will be of interest to the community. In the following, we respond to the detailed comments of the Reviewer.

Reviewer: 1. Fig. 1e is mislabeled as 1d.

Response: Thank you for spotting this error. We corrected Figure 1 accordingly.

Reviewer: 2. In Figs. 1a-b, 1e-f, the authors showed the schematics of the skyrmions with various helicities. However, it is not clear to me from the schematics how the spin arrangements are different in mixed helicities (Figs. 1e-f) from the fixed helicities (Figs. 1a-b). Since it is one of the main findings in this work, it would be beneficial to show clearly how the spin arrangements are different from the pure Bloch- or Neel-characters. (e.g., different color schemes for highlighting the spin orientations, higher pixel resolutions, etc).

Response: We thank the Reviewer for their comment. To improve the visibility of the spin arrangements of the different skyrmion types, we firstly improved the overall resolution of Figure 1, and revised Figures 1a, b, e, and f so that the reader can more easily inspect the spin structures. As noted by the Reviewer, Figs. 1a, b, e and f show the connection between the various skyrmion types and the spatial dependence of the helicity, with the helicity encoded by the relevant colourmap. Nonetheless, we agree with the Reviewer that it is important to show clear depictions of the magnetisation textures for the different skyrmion types, and how the magnetisation textures connect to the helicity distributions of the different skyrmions.

To this end, we extended Supplementary Note IX with a new Supplementary Figure 8 and associated text. With the new figure and its associated text (lines 338-377), we show in detail how the magnetisation textures and corresponding helicity distributions vary for the different skyrmions types in terms of presenting representative 1D-cuts along the directions of nearest- and next-nearest-neighbour directions of the skyrmion lattice. With this newly extended section of the Supplementary Information, we believe the connection between the magnetisation textures and spatial variation of helicities in Phases II and III is now more clearly visible, and more easily contrasted with standard Néel and Bloch skyrmions.

Reviewer: 3. Strictly speaking, the first term in eq.2 should be vorticity, as defined in previous literatures [1]. This vorticity will be equivalent to skyrmion number N_{sk} , only if the spin at the periphery is anti-parallel to that at the core. However, in the current manuscript, it is not clear to me that there is evidence of ruling out the possibilities for merons, which have spins confined in-plane at the periphery.

Response: We thank the Reviewer for this comment. To make the generalized description of skyrmion spin textures both clearer and more rigorous, we revised the relevant text in the Introduction on lines 46-56. We now introduce the definition of vorticity and its relation to the skyrmion number, and appropriate boundary conditions that allow us to directly relate these two quantities. According to the improved text, merons are ruled out as fitting in to the generalized description of the spin textures.

Reviewer: 4. At pg.5, line 81, the author claimed the “microscopic evidence for a hybrid helicity of skyrmions as an intrinsic property in bulk magnets has remained elusive.” However, skyrmions with mixed helicities have been reported more than 10 years ago [2].

Response: While we can agree with the Reviewers assessment in the broadest sense, the present work on EuNiGe_3 is focused on bulk samples in which thickness-dependent, i.e. dipolar interactions are considered to not play a key role in the formation of the hybrid spin textures. The Reference provided by the Reviewer includes observations of mixed helicity spin textures in a thin plate sample of an M-type hexaferrite system. In this study, sample thickness-dependent dipolar interactions are key for the stability of the hybrid spin textures, which means that their formation cannot be considered as bulk phenomenon. Nonetheless, on lines 85-88, we have both included this important reference as reference 27, and updated the text to stress the importance of the present work as one that concerns bulk, rather than low-dimensional samples.

Reviewer: 5. At pg.5, line 89, it would be beneficial for the readers to specify each of the Phases 1-4.

Response: We have followed the Reviewer’s suggestion and now specify the magnetic order in all four phases on lines 97-100.

Reviewer: 6. In Figs. 3e-h, the authors justify the ellipticity of SANS spots as elongation of skyrmions along various Q directions. However, in Fig. 3f, 3g & 3h, it appears to me that there are split spots. Please explain this feature.

Response: In the main text (lines 211 to 220) and Supplementary Note VI, we describe how the real-space elongation of skyrmions can only be relevant for Phase II. The elongation is inferred from the observation that the triple-Q vectors forming the Phase II skyrmion lattice do not form a perfect hexagon, but instead a distorted one. Therefore, a distortion of a multi-Q structure is only visible in Figure 3f, and not Figure 3g. The apparent elliptical shape of the individual diffraction spots in Figures 3e-h does not provide information about the real space distortion of skyrmions (or spirals in the cases of Phases I and IV), as arises naturally because of the SANS measurement geometry. To clarify the connection between the SANS data and the skyrmion distortion in real space as only relevant for Phase II, we updated the text on lines 217-218.

We also investigated the apparent split SANS spots in Figs. 3f-h and found that they result from the combined effects of weak diffraction intensity (due to high neutron absorption by Eu) and narrow diffraction (‘rocking’) curves. To collect SANS patterns efficiently across the phase diagram in a reasonable amount of neutron beamtime, we performed focused rocking scans, with detector measurements at selected sample rocking angles only near the Bragg conditions of the spots. We conclude that any split spots in Fig. 3f-h arise from a slight misplacement of rocking angle steps relative to the true Bragg angle during data collection. This minor technical shortcoming has no consequence on the subsequent SANS data analysis or interpretation. In the Methods section (lines

445-446), we provide more details of the measurement approach for collecting the data shown in Figure 3.

Reviewer: 7. In Fig. 3h, there are weak reflections in regions from $(q_x, q_y)=(-2, -2)$ to $(q_x, q_y)=(2, 0)$. Please explain such features.

Response: The weak reflections mentioned by the Reviewer correspond to statistical noise in the SANS data. In the mentioned detector regions, the associated intensity happens to be high enough relative to the defined intensity scale (colourbar), such that it becomes visible in the SANS images. To provide a more accurate representation of the background noise level relative to the diffraction spot intensities in the SANS data, we revised the intensity scale for Figures 3e-h.

Reviewer: 8. From symmetry perspective, the non-centrosymmetric crystal structure of EuNiGe₃ allows for the DMI to occur, which usually leads to skyrmions with same helicity. Using spin DFT and Monte Carlo simulations, the authors explained the hybrid helicity skyrmions originated from the frustration caused by Q-dependent DMI or anisotropic exchange interactions. If such frustration exists, why do the skyrmions still form lattices instead of aperiodic structures?

Response: The reason why the skyrmions form lattices instead of aperiodic structures might be attributed to the nature of the interactions involved. In the present itinerant magnet, the RKKY interaction plays an important role. Since the RKKY interaction is long-ranged and its magnitude shows maxima at specific wave vectors in momentum space, the alignment of the skyrmions with periodicities corresponding to those maxima lowers the energy. In other words, the present feature of skyrmion lattice formation will hold as long as the dominant contributions in the interaction are represented by specific momentum-resolved interactions.

Reviewer: References:

1. N. Nagaosa and Y. Tokura, Nature Nanotechnology 8, 899 (2013), “Topological properties and dynamics of magnetic skyrmions”.
2. X. Yu et al., PNAS 109, 8856 (2012), “Magnetic stripes and skyrmions with helicity reversals”.

REVIEWERS' COMMENTS

Reviewer #1 (Remarks to the Author):

The authors have addressed my questions and concerns in a very satisfactory way. The authors also respond the questions raised by the other reviewers. The current version of manuscript is ready for publication at Nature Communications.

One minor comment: the labels “ I, II, ..” for the different phase regions in Fig 1d are still too small to me.

Reviewer #2 (Remarks to the Author):

The authors addressed all my concerns. I thank Prof. Tokura for providing detailed replies.

Reviewer #3 (Remarks to the Author):

The authors have addressed my comments well. Therefore, I'm happy to recommend the paper for publication in Nature Communications.

Reviewer #1 (Remarks to the Author):

Reviewer: The authors have addressed my questions and concerns in a very satisfactory way. The authors also respond the questions raised by the other reviewers. The current version of manuscript is ready for publication at Nature Communications.

One minor comment: the labels “ I, II, ..” for the different phase regions in Fig 1d are still too small to me..

Response: We thank the Reviewer for their time reviewing our revised manuscript, and their approval for publication. In response to their minor comment, we have revised Fig. 1d so that the font size of the roman numeral labels matches that of the rest of the figure. We hope that with this small revision, the Reviewer will be satisfied that Fig. 1 is now of publication quality.

Reviewer #2 (Remarks to the Author):

Reviewer: The authors addressed all my concerns. I thank Prof. Tokura for providing detailed replies.

Response: We thank the Reviewer for their time reviewing our revised manuscript, and are pleased to read that our responses alleviated all of their earlier concerns.

Reviewer #3 (Remarks to the Author):

Reviewer: The authors have addressed my comments well. Therefore, I'm happy to recommend the paper for publication in Nature Communications.

Response: We thank the Reviewer for their positive assessment. We are happy to read that they recommend our paper for publication.